# Thermochemical nonequilibrium flow analysis in low enthalpy shock-tunnel facility

**Sanghoon Lee**[1], **Ikhyun Kim**[1], **Gisu Park**[1], **Jong Kook Lee**[2], **Jae Gang Kim**[3]*

**1** Department of Aerospace Engineering, Korea Advanced Institute of Science and Technology, Daejeon, Republic of Korea, **2** Agency for Defense Development, Daejeon, Republic of Korea, **3** Department of Aerospace System Engineering, Sejong University, Seoul, Republic of Korea

* jeagkim@sejong.ac.kr

**Data Availability Statement:** All relevant data are within the manuscript.

**Funding:** This study was conducted at the High-Speed Vehicle Research Center of KAIST with the support of the Defense Acquisition Program Administration (DAPA) and the Agency for Defense

## Abstract

A thermochemical nonequilibrium analysis was performed under the low enthalpy shock-tunnel flows. A quasi-one-dimensional flow calculation was employed by dividing the flow calculations into two parts, for the shock-tube and the Mach 6 nozzle. To describe the thermochemical nonequilibrium of the low enthalpy shock-tunnel flows, a three-temperature model is proposed. The three-temperature model treats the vibrational nonequilibrium of $O_2$ and NO separately from the single nonequilibrium energy mode of the previous two-temperature model. In the three-temperature model, electron-electronic energies and vibrational energy of $N_2$ are grouped as one energy mode, and vibrational energies of $O_2$, $O_2^+$, and NO are grouped as another energy mode. The results for the shock-tunnel flows calculated using the three-temperature model were then compared with existing experimental data and the results obtained from one- and two-temperature models, for various operating conditions of the K1 shock-tunnel facility. The results of the thermochemical nonequilibrium analysis of the low enthalpy shock-tunnel flows suggest that the nonequilibrium characteristics of $N_2$ and $O_2$ need to be treated separately. The vibrational relaxation of $O_2$ is much faster than that of $N_2$ in low enthalpy condition, and the dissociation rate of $O_2$ is manly influenced by the species vibrational temperature of $O_2$. The proposed three-temperature model is able to describe the thermochemical nonequilibrium characteristics of $N_2$ and $O_2$ behind the incident and reflected shock waves, and the rapid vibrational freezing of $N_2$ in nozzle expanding flows.

## Introduction

Hypersonic vehicles experience thermochemical nonequilibrium phenomena, which are induced by the hypersonic speed and the high enthalpy flow environment [1]. Real flight tests are critically important when designing hypersonic vehicles, to ensure they can endure such extreme flight conditions. However, real flight test is often impractical given the number of tests under various flight conditions that need to be performed, and their cost. As a result, hypersonic ground test facilities are widely used to supplement real flight tests. Among hypersonic ground facilities, shock-tunnels are commonly employed to simulate hypersonic flight

Development (No.UD140081CD). http://www.add.
re.kr/ The funders had no role in study design, data
collection and analysis, decision to publish, or
preparation of the manuscript.

**Competing interests:** The authors have declared
that no competing interests exist.

environments, due to their capability of simulating chemically reacting flows associated with hypersonic speed [2]. The shock-tunnel primarily consists of two parts, a shock-tube and a nozzle. In the shock-tube part, the gas in a driven tube is heated by incident and reflected shock waves which produce high-temperature flows. In the nozzle part, the heated gas expands drastically, while increasing the velocity of the flow to hypersonic speed. The test model is usually located at the nozzle exit and examined in hypersonic free-steam conditions.

To analyze the test results from a shock-tunnel facility, it is essential to rigorously diagnose the flowfield inside the facility. Especially, the real-gas effects of the flow at the nozzle exit, which the test model experiences, should be evaluated in detail, as the flow is chemically reactive and in a thermochemical nonequilibrium state [3]. Because real-gas effects can influence chemical reactions, the flow properties such as density, Mach number, and specific heat ratio often differ from theoretical predictions based on an assumption of equilibrium. Consequently, the shock shape around the sample as well as shock stand-off distance differ [4, 5]. In addition to this, rigorous understanding for the flowfield is also important in practical aspects. For example, the humid air could condense after the rapid expansion through the nozzle [6, 7]. In nozzle flow where the velocity gradient is large, the difference in time between the inertial restoration and temperature restoration is significant, which leads to thermodynamically nonequilibrium condensation [8, 9]. It has been reported that the liquid mass fraction, which would influence the measurement accuracy [10], is mainly determined by nonequilibrium condensation [11, 12]. Therefore, the accurate prediction of flowfield is important in analyzing the test results.

Unfortunately, it is difficult to measure all flow properties, because the flow duration is typically only a few milliseconds. It is also difficult to install instruments inside the facility without affecting the flow. Numerical simulations are therefore commonly performed to understand and diagnose the thermochemical nonequilibrium state inside the shock-tunnel facility and the hypersonic free-stream conditions at the nozzle exit. In numerical simulations, the three-dimensional flow calculations have an advantage of the performance analysis of the shock-tunnel facility [13]. However, the computational costs of the three-dimensional calculations are enormous to analyze the thermochemical nonequilibrium phenomena of the shock-tunnel flows. In the work by Nomelis et al. [14], the two-dimensional axi-symmetric calculations of the shock-tunnel flow were preformed, and it was compared with the results by the quasi-one-dimensional calculations and the measured data of the shock-tunnel facility. The comparisons confirmed that the quasi-one-dimensional calculations are acceptable to analyze the thermochemical nonequilibrium of the shock-tunnel flows. Hence, the thermochemical nonequilibrium analysis of the shock-tunnel flows has been widely performed by using the quasi-one-dimensional calculations [15]. In the present work, the Q1D calculations are employed to analyze the thermochemical nonequilibrium of the shock-tunnel flows for the numerical efficiency.

For the thermochemical nonequilibrium analysis, two-temperature model (2-T) [16] where translational energy and rotational energy are grouped as one energy mode and electronic-electron-vibrational energies are ground as another energy mode has been widely used. In high enthalpy flow conditions above 10 $MJ/kg$, it is sufficient to adopt the 2-T model to describe the thermochemical nonequilibrium state of the shock-tunnel flows, because the shock heated temperature is over 10,000 $K$, and the electron-electronic-vibrational temperature of the 2-T model can adequately capture the thermochemical nonequilibrium state in the shock-tunnel flows. However, in low enthalpy flow conditions, below 8 $MJ/kg$, the 2-T model is less accurate when analyzing thermochemical nonequilibrium flows in a shock-tunnel facility. In general, at enthalpy flow conditions below 8 $MJ/kg$, the shock heated temperature is less than 6,000 $K$, and the typical speed at the nozzle exit is below Mach number 6. In these low

enthalpy flow conditions, $N_2$ dissociation reactions rarely occur, and chemical reactions related to $O_2$ and NO are dominant. The chemical reactions related to $O_2$ and NO are mainly influenced by the vibrational temperatures of $O_2$ and NO species, not by the electron-electronic-vibrational temperature in the 2-T model. The vibrational temperatures of $O_2$ and NO species can be evaluated from vibrational relaxation times of $O_2$ and NO species, and in low enthalpy flows where the shock heated temperature is below 10,000 *K*, the vibrational relaxation times of $N_2$, $O_2$, and NO are quite different [17]. This makes it necessary to treat the species vibration temperature of $O_2$ and NO separately from the single electron-electronic-vibrational temperature in the 2-T model, when describing the thermochemical nonequilibrium state in low enthalpy shock-tunnel flows.

In the present work, a three-temperature (3-T) model is proposed for analyzing thermochemical nonequilibrium flows below 8 *MJ/kg* in a low enthalpy shock-tunnel facility. In the 3-T model, the nonequilibrium energy is divided into two energy modes. The electron translational energy, the electronic energies of atoms and molecules, and the vibrational energy of $N_2$ are grouped together as one nonequilibrium energy mode of the electron-electronic-vibrational energy. The other nonequilibrium energy mode is the species vibrational energy. In the species vibrational energy, the vibrational energies of $O_2$, NO, and $O_2^+$ are grouped and treated as another nonequilibrium energy mode. In the present study, a thermochemical nonequilibrium flow analysis of the K1 shock-tunnel facility [18, 19] was conducted using the proposed 3-T model. In the shock-tunnel flow calculation, the Q1D method was employed by dividing the flow calculations into two parts, the shock-tube and the Mach 6 nozzle. The results for the shock-tunnel flows calculated using the 3-T model were compared with existing experimental data [18, 20, 21] and the calculated results obtained from previous 1-T and 2-T models for various operating conditions in the K1 shock-tunnel facility.

## Thermochemical nonequilibrium model

### Chemical species and internal energies

In a typical shock-tunnel operation, dry air is used as a test gas in the driven tube to simulate the Earth's atmosphere, and He and Ar are mostly employed as the driver tube gas. Accordingly, a total of nine chemical species of N, O, He, Ar, $N_2$, $O_2$, NO, $O_2^+$, and $e^-$ were considered to calculate the shock-tunnel flows in the present work. The other chemical species of $N_2^+$, $NO^+$, $N^+$, $O^+$, $Ar^+$, and $He^+$ are ignored because they are rarely produced under the low enthalpy flow conditions in a shock-tunnel operating up to 8 *MJ/kg*.

For the electronic energy of N and O, the energy grouping model proposed by Hyun [22] was adopted, and the principal quantum number considered in this grouping model was limited to 10. Table 1 presents the electronic excitation levels of N and O adopted in the present energy grouping model. *n* is a principle quantum number, $E_{ex}$ is an electronic energy, and *g* is a multiplicity of the electronic state. In the present work, twenty and seventeen levels, respectively, were adopted in the electronic grouping model to evaluate the electronic energy of N and O. For Ar and He, only the electronic ground state was considered, because the driver gases of He and Ar exist at room temperature, and they rarely engage in the chemical reactions behind the incident and reflected shock waves.

Table 2 lists the electronic states of $N_2$, $O_2$, NO, and $O_2^+$ considered in the present work. The spectral data for each electronic state were obtained from the National Institute of Standards and Technology database [23], and the rotational and vibrational energies of each electronic state were calculated using the Dunham expansion model [24].

Fig 1 presents the calculated rotational and vibrational energies for $N_2$ and $O_2$ for the three low-lying electronic states. The total number of rotational and vibrational states were 8,870,

**Table 1. Electronic grouping energy levels [22] of N and O atoms.**

| N | | | | | | O | | | | | |
|---|---|---|---|---|---|---|---|---|---|---|---|
| $n$ | $E_{ex}$ ($cm^{-1}$) | $g$ | $n$ | $E_{ex}$ ($cm^{-1}$) | $g$ | $n$ | $E_{ex}$ ($cm^{-1}$) | $g$ | $n$ | $E_{ex}$ ($cm^{-1}$) | $g$ |
| 2 | 0.0 | 46 | 4 | 110315.0 | 90 | 2 | 78.0 | 9 | 5 | 103869.0 | 24 |
| 2 | 19228.0 | 10 | 4 | 110486.0 | 126 | 2 | 15868.0 | 5 | 5 | 105394.0 | 168 |
| 2 | 28840.0 | 6 | 5 | 111363.0 | 54 | 2 | 33792.0 | 1 | 6 | 106639.0 | 288 |
| 3 | 83337.0 | 12 | 5 | 112851.0 | 90 | 3 | 73768.0 | 5 | 7 | 107583.0 | 392 |
| 3 | 87488.0 | 18 | 5 | 112929.0 | 288 | 3 | 76795.0 | 3 | 8 | 108117.0 | 512 |
| 3 | 95276.0 | 36 | 6 | 114298.0 | 648 | 3 | 86629.0 | 15 | 9 | 108478.0 | 648 |
| 3 | 96793.0 | 18 | 7 | 115107.0 | 882 | 3 | 88631.0 | 9 | 10 | 108734.0 | 800 |
| 4 | 103862.0 | 18 | 8 | 115631.0 | 1152 | 4 | 95757.0 | 8 | | | |
| 3 | 104857.0 | 60 | 9 | 115991.0 | 1458 | 3 | 97745.0 | 40 | | | |
| 3 | 104902.0 | 30 | 10 | 116248.0 | 1800 | 4 | 99313.0 | 24 | | | |

3,359, and 3,914 for $N_2$($X^1\Sigma_g^+$, $A^3\Sigma_u^+$, and $B^3\Pi_g$), and 5,117, 3,854, and 3,221 for $O_2$($X^3\Sigma_g^-$, $a^1\Delta_g$, and $b^1\Sigma_g^+$), respectively. For the electronic ground $N_2$($X^1\Sigma_g^+$) state, the maximum vibrational state $v_{max}$ was 60, and the maximum rotational state $J_{max}$ was 266. For the electronic ground $O_2$($X^3\Sigma_g^-$) state, $v_{max}$ and $J_{max}$ were 36 and 222, respectively. In the present work, all of the rotational and vibrational energies for each electronic state in Table 2 were adopted to determine the average electronic, rotational, and vibrational energies of atoms and molecules, and the average energies were determined using the partition function relations specified by the nonequilibrium temperatures [24]. Then, the specific heats $c_v$ and $c_p$ were directly determined from the average energies of the atoms and molecules.

## Three-temperature model

To describe the thermochemical nonequilibrium phenomena in low enthalpy shock-tunnel flows, the present work introduces a 3-T model. In low enthalpy shock-tunnel flows below 8 $MJ/kg$, the chemical reactions related to $N_2$ occur slowly, and chemical reactions related to $O_2$ and NO occur more rapidly than the chemical reactions related to $N_2$ [25]. The chemical reactions related to $O_2$ and NO are controlled by the species vibrational temperatures of $O_2$ and NO.

Fig 2 presents a schematic diagram of the nonequilibrium energies and temperatures in the 3-T model. In the 3-T model, the translational and rotational energies are treated as one equilibrium energy mode, specified by trans-rotational temperature $T_{tr}$, because the rotational energy rapidly equilibrates with the translational energy. Unlike the 2-T model, the nonequilibrium energy mode of the electron–electronic-vibrational energy is divided into two nonequilibrium energy modes: the electron–electronic-vibrational energy and the species vibrational energy. In the electron-electronic-vibrational energy, the electron translational

**Table 2. Electronic states of $N_2$, $O_2$, NO, and $O_2^+$.**

| Species | Electronic States |
|---|---|
| $N_2$ | $X^1\Sigma_g^+$, $A^3\Sigma_u^+$, $B^3\Pi_g$, $W^3\Delta_u$, $a'^1\Pi_g$, $w^3\Delta_u$, $A'^5\Sigma_g^+$, $G^3\Delta_g$, $C^3\Pi_u$, $b^1\Pi_u$, $b^1\Pi_u$, $b'^1\Sigma_u^+$ |
| $O_2$ | $X^3\Sigma_g^-$, $a^1\Delta_g$, $b^1\Sigma_g^+$, $c^1\Sigma_u^-$, $A'^3\Delta_u$, $A^3\Sigma_u^+$, $B^3\Sigma_u^-$ |
| NO | $X^2\Pi$, $a^4\Pi$, $A^2\Sigma^+$, $B^2\Pi$, $b^4\Sigma^-$, $C^2\Pi$, $D^2\Sigma^+$, $B'^2\Delta$, $E^2\Sigma^+$, $F^2\Delta$, $H^2\Sigma^+$, $H'^2\Pi$ |
| $O_2^+$ | $X^2\Pi_g$, $a^4\Pi_u$, $A^2\Pi_u$ |

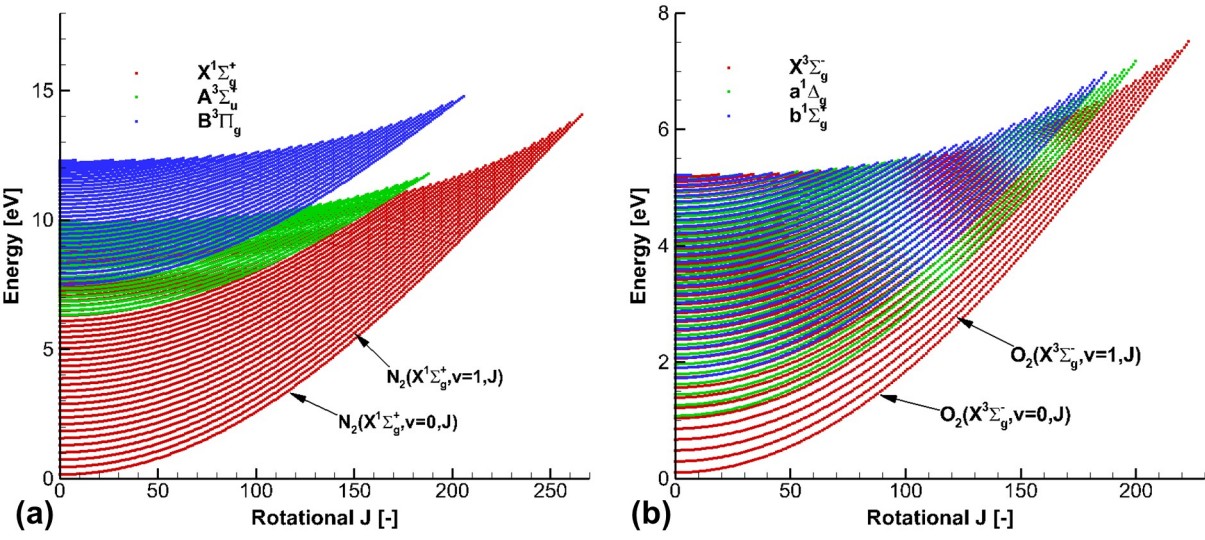

**Fig 1. Rotational and vibrational energies for the three low-lying electronic states.** (a) $N_2$ and (b) $O_2$.

energy, the electronic energies of atoms and molecules, and the vibrational energy of $N_2$ are grouped as one energy mode, and are represented by a Boltzmann distribution specified by the electron–electronic-vibrational temperature $T_{eev}$. In the species vibrational energy, the vibrational energies of $O_2$, NO, and $O_2^+$ are grouped as one species vibrational energy, and are described by a Boltzmann distribution specified by the species vibrational temperature $T_v$. This is because nonequilibrium properties of NO, e.g. relaxation times for vibrational-electron energy exchange and vibrational-translational energy exchange, are similar to those of $O_2$ [26] and also, the formation of NO is not significantly influenced by the species vibrational temperature [27]. By grouping the vibrational energies of $O_2$ and NO, the computational cost is reduced. Then, the chemical reactions related to $N_2$ and electrons are controlled by the electron-electronic-vibrational temperature $T_{eev}$, and the chemical reactions related to $O_2$ are influenced by the species vibrational temperature $T_v$. The use of a 3-T model enables efficient

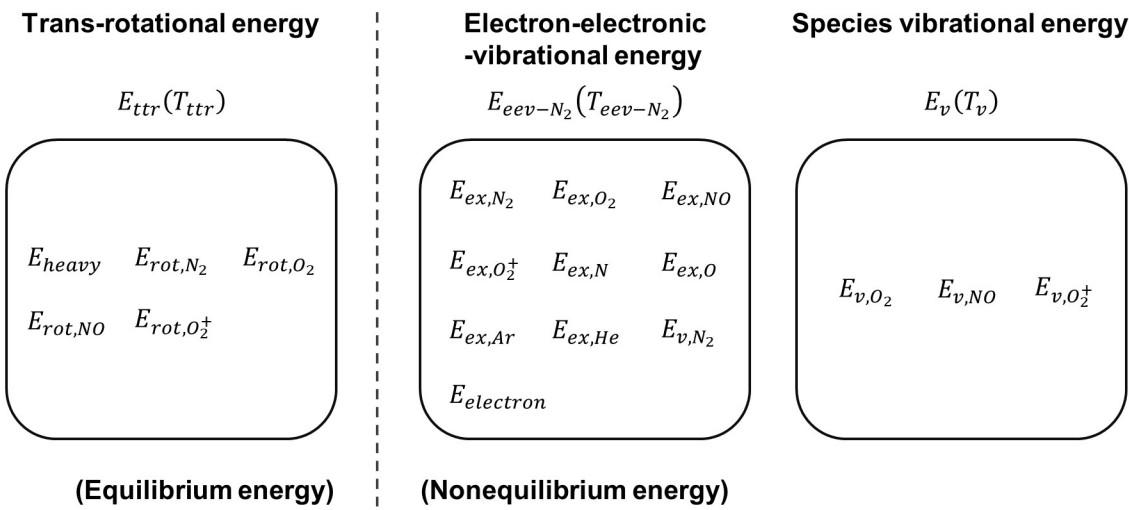

**Fig 2. Schematic diagram of nonequilibrium energies and temperature in the 3-T model.**

calculation of the thermochemical nonequilibrium flow, including the different vibration characteristics of $N_2$ and $O_2$ and related chemical reactions in the low enthalpy shock-tunnel operating environment.

In order to describe the energy transferred among the three-energy modes of the 3-T model, the vibrational-to-translational (V-T), vibrational-to-electron (V-e), vibrational-to-vibrational (V-V), and electron-to-translational (e-T) energy transfers need to be considered.

**V-T energy exchange.** In the present work, the Landau-Teller model [17] is employed to describe the V-T energy exchange as follows:

$$\dot{w}_{eev}^{V-T} = \rho_{N_2} \frac{\bar{e}_{v,N_2}(T_{tr}) - \bar{e}_{v,N_2}(T_{eev})}{\bar{\tau}_{v,N_2}}, \tag{1}$$

$$\dot{w}_v^{V-T} = \sum_{s=O_2,NO,O_2^+} \rho_s \frac{\bar{e}_{v,s}(T_{tr}) - \bar{e}_{v,s}(T_{eev})}{\bar{\tau}_{v,s}}, \tag{2}$$

where $\dot{w}_{eev}^{V-T}$ and $\dot{w}_v^{V-T}$ are the V-T energy exchange rate of the electron-electronic-vibrational and species vibrational energy modes, respectively. $\rho$ is the density, $\bar{e}_v$ is the average vibrational energy per mass, and $\bar{\tau}_v$ is the average vibrational relaxation time in the V-T energy transferring. The average vibrational relaxation time for species $i$ $\bar{\tau}_{v,i}$ is determined from the vibrational relaxation time for the collision between species $i$ and species $j$ $\tau_{v,ij}$ as

$$\frac{1}{\bar{\tau}_{v,i}} = \frac{\sum_j \rho_j/M_j \tau_{v,ij}}{\sum_j \rho_j/M_j}, \tag{3}$$

where $M_i$ is a molar mass for species $i$. The vibrational relaxation time for the collision between species $i$ and species $j$ $\tau_{v,ij}$ can be determined using the Millikan-White (M-W) formula [17]:

$$p\tau_{v,ij} = \exp\left[A_{ij}(T_{tr}^{-1/3} - B_{ij}) - 18.42\right]. \tag{4}$$

where $p$ is pressure, $A_{ij}$ and $B_{ij}$ are the modified M-W parameters. In the present work, the M-W parameters of the nine-species for the low enthalpy shock-tube flows were obtained from the work by Park et al. [28, 29], Millikan and White [17], and Kim et al. [25, 30].

**V-e energy exchange.** In describing the V-e energy transferring by the collisions with electrons, the model proposed by Lee [31] was adopted in the present work:

$$\dot{w}_{eev}^{V-e} = -\dot{w}_v^{V-e}, \tag{5}$$

$$\dot{w}_v^{V-e} = \sum_{s=O_2,NO,O_2^+} \frac{\bar{e}_{v,s}(T_{eev}) - \bar{e}_{v,s}(T_v)}{\tau_{e,s}}, \tag{6}$$

where $\dot{w}_{eev}^{V-e}$ and $\dot{w}_v^{V-e}$ are the V-e energy exchange rate of the electron-electronic-vibrational and species vibrational energy modes, respectively. $\tau_{e,s}$ is the relaxation time in the V-e energy transferring. The relaxation time $\tau_e$ for $O_2$ and NO were obtained using Lee's work [31]. Since the V-e energy coupling of $O_2$ and NO is slower than that of $N_2$, the relaxation time $\tau_e$ for $O_2$ and NO were adopted by multiplying the value for $N_2$ by 300, as proposed by Park and Lee [32].

**V-V energy exchange.** The V-V energy exchange from the collisions between the diatomic molecules was given by Candler and MacCormack [33]:

$$\dot{w}_{eev}^{V-V} = \sum_{s=O_2,NO,O_2^+} P_{N_2-s}^{V-V} Z_{N_2-s} [\bar{e}_{v,s}(T_v) - \bar{e}_{v,N_2}(T_{eev})], \tag{7}$$

$$\dot{w}_{v}^{V-V} = \sum_{s=O_2,NO,O_2^+} P_{N_2-s}^{V-V} Z_{N_2-s} [\bar{e}_{v,N_2}(T_{eev}) - \bar{e}_{v,s}(T_v)], \tag{8}$$

where $P_{N_2-s}^{V-V}$ is the collision probability for V-V energy transferring between $N_2$ and species $s$, and $Z$ is the collision number. For the collisions between species $i$ and $j$, it is given by

$$Z_{i,j} = \left( \frac{8kT_{tr}}{\pi \mu_{i,j}} \sigma_{i,j} n_i n_j \right), \tag{9}$$

where $\dot{w}_{eev}^{V-V}$ and $\dot{w}_{v}^{V-V}$ are the V-V energy exchange rate of the electron-electronic-vibrational and species vibrational energy modes, respectively. $k$ is the Boltzmann constant, $n_i$ is the number density of species $i$, and $\mu_{i,j}$ is the reduced mass between species $i$ and $j$. The elastic collision cross section $\sigma_{i,j}$ was taken to be equal to $1.0 \times 10^{-19}$ $m^2$ as proposed by Park [32]. The collision property $P_{N_2-s}^{V-V}$ for V-V energy transfer was determined from existing experimental data [34] as:

$$P_{N_2-O_2}^{V-V} = 3.0 \times 10^{-6}(T_{tr}/1,000)^{2.87}, \tag{10}$$

$$P_{N_2-NO}^{V-V} = 5.5 \times 10^{-5}(T_{tr}/1,000)^{2.32}. \tag{11}$$

The probability of collision $P_{N_2-s}^{V-V}$ between $N_2$ and $O_2^+$ was treated as the same as the collisions between $N_2$ and $O_2$.

**e-T energy exchange.** In the present work, the energy exchange between the electron and heavy particle translational energies [35] is expressed by using the effective collision frequency $\nu_e$ as

$$\dot{w}_{eev}^{e-T} = 2k(T_{tr} - T_{eev})n_e M_e \Sigma_i \frac{\nu_{e,i}}{M_i}. \tag{12}$$

where $\dot{w}_{eev}^{e-T}$ is the e-T energy exchange rate of the electron-electronic-vibrational energy mode, and the subscript $e$ is electron. The effective collision frequency $\nu_{e,i}$ can be divided into two types of collisions: electron-neutral and electron-ion. For collisions between the electron $e$ and neutral species $i$, the cross section $\sigma_{e,i}$ of the effective electron energy exchange [36] is used:

$$\nu_{e,i} = n_i \sigma_{e,i} \sqrt{\frac{8kT_{eev}}{\pi m_e}}, \tag{13}$$

where $m_e$ is a mass of electron per particle. The cross section $\sigma_{e,i}$ was obtained from the work by Gupta et al. [37]. The collision between the electron $e$ and charged species of $O_2^+$ [36] was evaluated as:

$$\nu_{e,O_2^+} = \frac{8}{3} \left( \frac{\pi}{m_e} \right)^{0.5} n_{O_2^+} e_e^4 \frac{1}{2kT_{eev}^{2/3}} \log \left( \frac{k^3 T_{eev}^3}{\pi n_e e_e^6} \right), \tag{14}$$

where $e_e$ is the magnitude of the electronic charge.

## Chemical reaction

In the present work, to describe the chemical reactions in the low enthalpy shock-tunnel flows the dissociation, exchange reaction, and associative ionization of the nine chemical species were considered. In the 3-T model, the chemical reactions are calculated using an Arrhenius type equation with the geometrically averaged temperature concept [16, 28, 29, 32]. Then the forward rate coefficients $k_f$ of the Arrhenius type equation can be expressed as

$$k_f = C_f T_f^{\eta} \exp\left(-\frac{\theta}{T_f}\right). \tag{15}$$

In Table 3, the Arrhenius parameters $C_f$, $\eta$, and $\theta$ are presented for the dissociation, exchange reaction, and associative ionization of the nine-chemical species. The parameters

**Table 3. Arrhenius parameters for the nine-chemical species in the low enthalpy shock-tunnel flows [25, 28–30, 38].**

| Reaction | $M$ | $C_f$ | $\eta$ | $\theta$ | $T_f$ | $T_r$ |
|---|---|---|---|---|---|---|
| Dissociation | | | | | | |
| $N_2+M \rightleftharpoons N+N+M$ | N | $3.0 \times 10^{22}$ | −1.6 | 113, 200 | $\sqrt{T_{tr}T_{eev}}$ | $T_{tr}$ |
| | O | $3.0 \times 10^{22}$ | −1.6 | 113, 200 | $\sqrt{T_{tr}T_{eev}}$ | $T_{tr}$ |
| | He | $3.0 \times 10^{22}$ | −1.6 | 113, 200 | $\sqrt{T_{tr}T_{eev}}$ | $T_{tr}$ |
| | Ar | $2.0 \times 10^{21}$ | −1.5 | 113, 200 | $\sqrt{T_{tr}T_{eev}}$ | $T_{tr}$ |
| | $N_2$ | $7.0 \times 10^{21}$ | −1.6 | 113, 200 | $\sqrt{T_{tr}T_{eev}}$ | $T_{tr}$ |
| | $O_2$ | $7.0 \times 10^{21}$ | −1.6 | 113, 200 | $\sqrt{T_{tr}T_{eev}}$ | $T_{tr}$ |
| | NO | $7.0 \times 10^{21}$ | −1.6 | 113, 200 | $\sqrt{T_{tr}T_{eev}}$ | $T_{tr}$ |
| | $O_2^+$ | $7.0 \times 10^{21}$ | −1.6 | 113, 200 | $\sqrt{T_{tr}T_{eev}}$ | $T_{tr}$ |
| $O_2+M \rightleftharpoons O+O+M$ | N | $1.0 \times 10^{22}$ | −1.5 | 59, 500 | $\sqrt{T_{tr}T_v}$ | $T_{tr}$ |
| | O | $3.0 \times 10^{21}$ | −1.5 | 59, 500 | $\sqrt{T_{tr}T_v}$ | $T_{tr}$ |
| | He | $1.0 \times 10^{22}$ | −1.5 | 59, 500 | $\sqrt{T_{tr}T_v}$ | $T_{tr}$ |
| | Ar | $1.8 \times 10^{18}$ | −1.0 | 59, 500 | $\sqrt{T_{tr}T_v}$ | $T_{tr}$ |
| | $N_2$ | $2.0 \times 10^{21}$ | −1.5 | 59, 500 | $\sqrt{T_{tr}T_v}$ | $T_{tr}$ |
| | $O_2$ | $1.117 \times 10^{25}$ | −2.6 | 59, 500 | $\sqrt{T_{tr}T_v}$ | $T_{tr}$ |
| | NO | $2.0 \times 10^{21}$ | −1.5 | 59, 500 | $\sqrt{T_{tr}T_v}$ | $T_{tr}$ |
| | $O_2^+$ | $1.117 \times 10^{25}$ | −2.6 | 59, 500 | $\sqrt{T_{tr}T_v}$ | $T_{tr}$ |
| $NO+M \rightleftharpoons N+O+M$ | N | $1.1 \times 10^{17}$ | 0.0 | 75, 500 | $\sqrt{T_{tr}T_v}$ | $T_{tr}$ |
| | O | $1.1 \times 10^{17}$ | 0.0 | 75, 500 | $\sqrt{T_{tr}T_v}$ | $T_{tr}$ |
| | He | $1.1 \times 10^{17}$ | 0.0 | 75, 500 | $\sqrt{T_{tr}T_v}$ | $T_{tr}$ |
| | Ar | $5.5 \times 10^{20}$ | −1.5 | 75, 500 | $\sqrt{T_{tr}T_v}$ | $T_{tr}$ |
| | $N_2$ | $5.0 \times 10^{15}$ | 0.0 | 75, 500 | $\sqrt{T_{tr}T_v}$ | $T_{tr}$ |
| | $O_2$ | $5.0 \times 10^{15}$ | 0.0 | 75, 500 | $\sqrt{T_{tr}T_v}$ | $T_{tr}$ |
| | NO | $1.1 \times 10^{17}$ | 0.0 | 75, 500 | $\sqrt{T_{tr}T_v}$ | $T_{tr}$ |
| | $O_2^+$ | $5.0 \times 10^{15}$ | 0.0 | 75, 500 | $\sqrt{T_{tr}T_v}$ | $T_{tr}$ |
| Exchange reaction | | | | | | |
| $N_2+O \rightleftharpoons NO+N$ | | $6.4 \times 10^{17}$ | −1.0 | −38, 400 | $T_{tr}$ | $T_{tr}$ |
| $NO+O \rightleftharpoons O_2+N$ | | $8.4 \times 10^{12}$ | 0.0 | −19, 450 | $T_{tr}$ | $T_{tr}$ |
| Associative ionization | | | | | | |
| $O+O \rightleftharpoons O_2^+ + e^-$ | | $7.1 \times 10^2$ | 2.7 | −80, 600 | $T_{tr}$ | $T_{eev}$ |

were obtained from the work by Park et al. [28, 29] and Kim et al. [25, 30], and the reference data by Baulch et al [38]. In the present work, the charged exchange reaction, ionization, and associative ionization except $O_2^+$ are not considered because in low enthalpy flows below 8 *MJ/kg*, such types of reactions are ignorable [25]. In the 3-T model, the $N_2$ dissociation is controlled by the electron-electronic-vibrational temperature $T_{eev}$, and the $O_2$ and NO dissociation reactions are controlled by the species vibrational temperature $T_v$, as shown in Table 3.

In reverse reactions, an equilibrium constant $K_e$ is employed to determine the backward rate coefficients $k_b$ as

$$k_b = \frac{k_f(T_{tr})}{K_e(T_{tr})}. \tag{16}$$

In the present work, the equilibrium constants $K_e$ for the dissociation, exchange reaction, and associated ionization for the nine-chemical species were obtained from the work by Kim [39] and Kim et al. [25].

In the species mass conservation, the mass rate of *s*-species production $\dot{\omega}_s$ is expressed as follows:

$$\dot{\omega}_s = M_s \sum_{k=1}^{N_k} (\beta_{s,k} - \alpha_{s,k})(R_{f,k} - R_{b,k}), \tag{17}$$

where $N_k$ is the number of reactions, $\alpha_{s,k}$ and $\beta_{s,k}$ are the stoichiometric coefficients for the reactants and products in the *k*-th reaction. Then, the forward and reverse reaction rates, $R_{f,k}$ and $R_{b,k}$, of the *k*-th reaction are defined by

$$R_{f,k} = 1000 \left[ k_{f,k} \prod_{s=1}^{all} (0.001 \rho_s / M_s)^{\alpha_{s_k}} \right], \tag{18}$$

$$R_{b,k} = 1000 \left[ k_{b,k} \prod_{s=1}^{all} (0.001 \rho_s / M_s)^{\beta_{s_k}} \right]. \tag{19}$$

In order to describe energy conservation due to dissociation, the preferential dissociation model [16, 29, 32] is employed. In the preferential dissociation model, the losses of electron-electronic-vibrational energy $\dot{w}_{eev}^D$ and species vibrational energy $\dot{w}_v^D$ due to dissociation can be defined as follows:

$$\dot{w}_{eev}^D = \sum_{s=N_2} \dot{\omega}_s^D (\psi_{v,s} E_s^D + \bar{e}_{ex,s}), \tag{20}$$

$$\dot{w}_v^D = \sum_{s=O_2, NO} \dot{\omega}_s^D (\psi_{v,s} E_s^D + \bar{e}_{ex,s}), \tag{21}$$

where $\dot{\omega}_s^D$ is the mass rate of production by dissociation, $E_s^D$ is the dissociation potential of species *s* per unit mass, $\bar{e}_{ex,s}$ is an average electronic energy of species *s*, and $\psi_{v,s}$ is the vibrational energy loss ratio of molecule *s* due to dissociation. In the work by Park [28], the value of 0.3 is proposed for $\psi_{v,s}$ in the atmospheric gas species.

Consequently, the overall rates of change in the electron-electronic-vibrational and species vibrational energies by the V-T, V-e, V-V, and e-T energy exchanges and chemical reactions

are determined as follows:

$$\dot{w}_{eev} = \dot{w}_{eev}^{V-T} + \dot{w}_{eev}^{e-V} + \dot{w}_{eev}^{V-V} + \dot{w}_{eev}^{e-T} + \dot{w}_{eev}^{D},$$ (22)

$$\dot{w}_{v} = \dot{w}_{v}^{V-T} + \dot{w}_{v}^{e-V} + \dot{w}_{v}^{V-V} + \dot{w}_{v}^{D}.$$ (23)

## Quasi-one-dimensional flow solver

The governing equations of Q1D flows for the 3-T model were constructed using five-types of equations; the species conservation, momentum conservation, electron-electronic-vibrational energy conservation, species vibrational energy conservation, and total energy conservation equations. The conservation equations for the Q1D system can be expressed as follows:

$$\frac{\partial Q}{\partial t} + \frac{\partial E}{\partial x} = W,$$ (24)

where $t$ is time and $x$ is position. The conserved variables vector $Q$, the corresponding total flux vector $E$, and the source terms vector $W$ can be represented as follows:

$$Q = \begin{bmatrix} \rho_s \\ \vdots \\ \rho u \\ E_{eev} \\ E_v \\ E_{tot} \end{bmatrix}, \; E = \begin{bmatrix} \rho_s u \\ \vdots \\ \rho u^2 + p \\ E_{eev} u \\ E_v u \\ E_{tot} u \end{bmatrix}, \; W = \begin{bmatrix} \dot{w}_s \\ \vdots \\ \dot{w}_{visc} \\ \dot{w}_{eev} \\ \dot{w}_v \\ 0 \end{bmatrix},$$ (25)

where $p$ is pressure, $u$ is velocity, and $E_{eev}$ and $E_v$ is the electron-electronic-vibrational and species vibrational energies per unit volume, respectively. In Q1D calculations, the viscous effects are modelled as the viscous source term $\dot{w}_{visc}$ in the momentum conservation equation. The wall-skin force of the shock-tunnel flows are calculated as follows:

$$\dot{w}_{visc} = \frac{p}{A}\frac{dA}{dx} - \frac{\pi d\tau_0}{A},$$ (26)

where $A$ is area. In Eq 26, the area change and mixture viscosity are considered. For the determination of shear stress $\tau_0$, the Darcy-Weisbach friction factor $f$ [15, 40, 41] for the pipe flow has been widely used for Q1D calculations [13, 15], and can be represented as follows:

$$\tau_0 = \frac{-\rho f u|u|}{8}.$$ (27)

The Darcy-Weisbach friction factor $f$ can be evaluated by Reynolds number [42, 43]. In this study, the Darcy-Weisbach friction factor $f$ for turbulent regime is obtained from Ref [44] for high Reynolds number up to $10^5$ as follows:

$$f = \frac{1}{\Lambda}[1.8 \log_{10} Re - 1.5147]^{-2},$$ (28)

where $\Lambda$ is compressibility factor, and the effective Reynolds number is defined as

$$Re = \frac{\rho^* u d}{\mu(T^*)}, \tag{29}$$

where $\mu$ is the mixture viscosity coefficient and $d$ is diameter. The reference density $\rho^*$ and temperature $T^*$ are determined by using the Eckert reference values [45] as,

$$T^* = T + 0.5(T_w - T) + 0.22(T_{aw} - T), \tag{30}$$

$$\rho^* = \frac{\rho T}{T^*}. \tag{31}$$

The adiabatic wall temperature $T_{aw}$ is evaluated by using the compressibility factor $\Lambda$, which compensates for the high-speed flow and is defined as

$$T_{aw} = \Lambda T, \tag{32}$$

$$\Lambda = 1 + \frac{\gamma - 1}{2} \Omega M^2, \tag{33}$$

where $\gamma$ is specific heat ratio and $M$ is Mach number. The recovery factor $\Omega$ is determined from Prandtl number $Pr$, as $Pr^{1/2}$ for laminar flow and $Pr^{1/3}$ for turbulent flow. The viscosity coefficient of the mixture gas $\mu$ is calculated using Gupta's mixing rule [46],

$$\mu(T^*) = \Sigma_i \frac{m_i n_i}{\Sigma_j n_j \Delta_{i,j}^{(2)}(T^*)}, \tag{34}$$

$$\Delta_{i,j}^{(2)}(T^*) = \frac{16}{5}\left[\frac{2M_i M_j}{\pi \bar{R} T(M_i + M_j)}\right]^{1/2} \pi \Omega_{i,j}^{(2,2)(T^*)}, \tag{35}$$

where $m_i$ is a mass of species $i$ per particle and $\bar{R}$ is the universal gas constant. The collision integrals $\Omega_{i,j}^{(2,2)}$ of the nine-chemical species in the shock-tunnel flows were obtained from the work by Kim et al. [47] and Wright et al. [48].

The Q1D equations in Eq 24 were calculated using the finite volume method. The flow variables are stored at the cell centers, and the quantities $Q$ at the cell faces were computed from the solution reconstruction based on the cell-centered data. The modified Steger-Warming flux vector splitting method [49] was employed in treating the reconstruction of the inviscid flux. The speed of sound $a$ for determining the characteristics of the inviscid flux was evaluated from the equation as

$$\rho a^2 = \sum_{s=1}^{all} \rho_s \frac{\partial p}{\partial \rho_s} + 2\rho u \frac{\partial p}{\partial \rho u} + E_{eev} \frac{\partial p}{\partial E_{eev}} + E_v \frac{\partial p}{\partial E_v} + \left(E_{tot} + p + u\frac{\partial E_{tot}}{\partial u}\right)\frac{\partial p}{\partial E_{tot}}. \tag{36}$$

In the present work, the fourth-order Runge-Kutta-Gill method was employed to perform an explicit time integration of the conservative equations. To achieve numerical efficiency, the flow solver was parallelized by using the MPI library.

## Verification and validation

Fig 3 presents the shock-tube and nozzle flow calculations used to validate the present calculations, by comparison with existing experimental data [50, 51]. In the shock-tube experiments

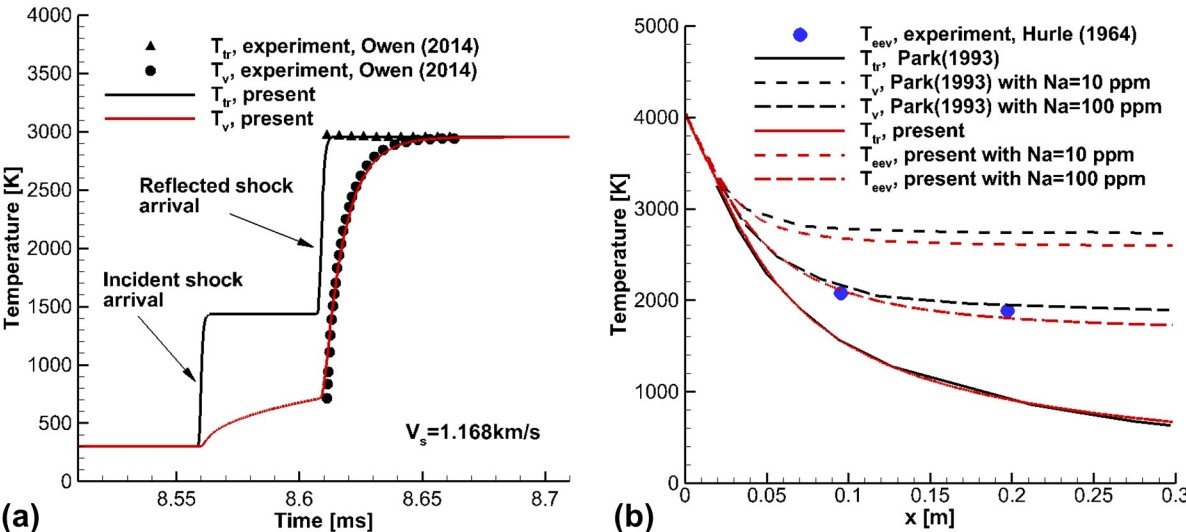

**Fig 3. Comparison of temperatures from the present calculations and the experimental data [50, 51].** (a) Temperature at the observation point of the shock-tube flows, (b) Temperature in nozzle expanding flows.

[50], the vibrational temperature $T_v$ was modeled based on the UV absorption spectrum of the Schumann-Runge band, in the wavelength range of 130–270 $nm$. A 2% $O_2$-98% Ar mixture gas was adopted as the test gas, and the pressure of the driven tube as $p_1 = 7.12$ $Torr$. The measured incident shock velocity was 1.174 $km/s$. The shock-tube had a driver section length of 3,700 $mm$ and a driven section length of 10,000 $mm$ with a diameter of 152.4 $mm$. The observation point was 2.0 $cm$ away from the end-wall. In the present work, the shock-tube flow calculation was carried out with a mesh of 5,000 elements, and the CFL number was set at a constant of 0.01.

In Fig 3(a), the incident shock wave reached the observation point 8.56 $ms$ after the diaphragm ruptured, and the reflected shock wave reached the point 0.05 $ms$ after the arrival of the incident shock wave. Accordingly, the calculated incident shock speed is 1.168 $km/s$. The calculated trans-rotational temperature $T_{tr}$ behind the reflected shock wave is 2,964 $K$, and this value is in good agreement with the measured equilibrium temperature. It was observed that the measured vibrational temperature behind the reflected shock wave was accurately reproduced by the present calculation.

Fig 3(b) shows a comparison between the calculated electron-electronic-vibrational temperature $T_{eev}$ in the nozzle expanding flows and the experimental data [51] and the results calculated by Park [32]. In experiments performed by Hurle et al. [51], the spectrum-line reversal method with Na was employed to measure the electron-electronic-vibrational temperature $T_{eev}$. The electronic temperature of Na was modeled based on the emission spectra of Na. The measured electronic temperature was then treated as the same as the vibrational temperature of $N_2$, because the interchange of $N_2$ vibrational energy and electronic energy of a metallic particle is rapid and highly effective, so that electronic temperature of Na easily equilibrates with the $N_2$ vibrational temperature [31]. As seed species can enhance the rate of vibrational relaxation process [52], Park [32] employed the physically fastest vibrational relaxation time for the collision between $N_2$ and Na and calculated the nozzle flow. In that calculation, the concentration of Na was varied between 10 and 100 ppm by volume. In the present work, the nozzle flow calculations were conducted using the initial conditions of $p_0 = 5$ $atm$ and $T_0 = 4,500$ $K$. The nozzle shape and the vibrational relaxation time for Na collisions were obtained from the

work by Park [32]. A mesh of 1,000 elements was used in the nozzle flow calculations, and the CFL was set as a constant of 0.01. In Fig 3(b), the trans-rotational temperature $T_{tr}$ decreases drastically, and the electron-electronic-vibrational temperature $T_{eev}$ is frozen in the nozzle expanding flows. In comparison with the experimental data, the calculated electron-electronic-vibrational temperature $T_{eev}$ agreed well with the measured temperature for a Na concentration of 100 ppm. In addition, the trans-rotational temperature $T_{tr}$ and electron-electronic-vibrational temperature $T_{eev}$ in present calculation were similar to the calculation results of Park [32].

## Thermochemical nonequilibrium analysis in the K1 shock-tunnel facility

### K1 shock-tunnel facility

Fig 4 presents the experimental setup of the K1 shock-tunnel [18, 19], installed at KAIST. The K1 shock-tunnel consists of four parts, the shock-tube, Mach 6 nozzle, test section, and dump tank. The facility can operate in shock-tube mode or as a reflected shock-tunnel, depending on the experimental requirements [53]. In the shock-tube part, the length of the driver and the driven sections are 865 $mm$ and 3,290 $mm$, respectively. The driver section has a diameter of 67.5 $mm$, and the driven section has a diameter of 47.5 $mm$. Two 0.35-$mm$-thick polyethylene diaphragms, which are placed at both ends of the transition section, separate the driver from the shock-tube. A secondary 0.04-$mm$-thick polyethylene diaphragm is used to separate the shock-tube from the nozzle and the test section. In shock-tunnel operating mode, depending on the test condition, the driver tube can be expanded to a length of 2,400 $mm$ using double diaphragms. Considering the connecting section between the shock-tube and the nozzle, the length of the driven section is enlarged to 3,360 $mm$. Three PCB piezoelectric pressure transducers are installed to measure both the static and reservoir pressures, and the corresponding shock speed. In Fig 4, pt1, pt2, and pt3 denote the flush-mounted transducers in the wall of the driven tube used for pressure measurement. The pitot pressure is measured using a flush-mounted flat-face model located approximately at the nozzle-exit center, with a PCB transducer. A 0.01-$mm$-thick polyvinyl-chloride film is attached in front of the transducer to protect the transducer from particle strikes which may be generated from the diaphragms. The K1 shock-tunnel is designed to simulate an enthalpy range of 1.0-8.0 $MJ/kg$ in dry-air gas. A detailed explanation of the shock-tunnel's operation can be found in Ref. [18] and Ref. [19].

The sample shock-tunnel flow calculations were conducted under initial conditions of $p_4$ = 3.1 $MPa$ and $T_4$ = 300 $K$ in the driver section, and $p_1$ = 40 $kPa$ and $T_1$ = 300 $K$ in the driven section, which were the reference operating conditions of the K1 shock-tunnel to produce Mach 6 flows at the nozzle exit. He and dry air were employed as the driver and test gases, respectively.

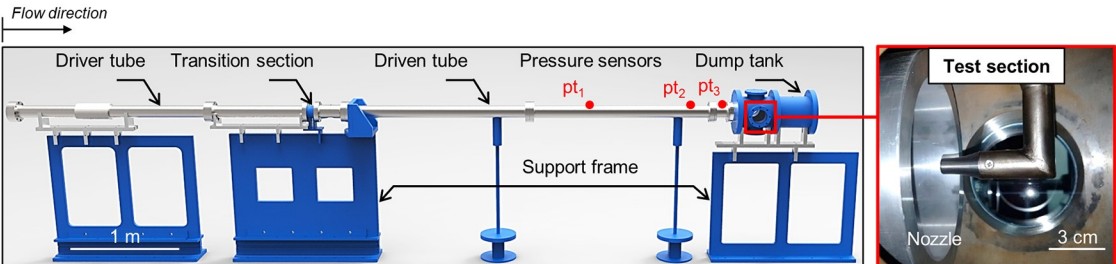

**Fig 4. Configuration of K1 shock-tunnel facility.**

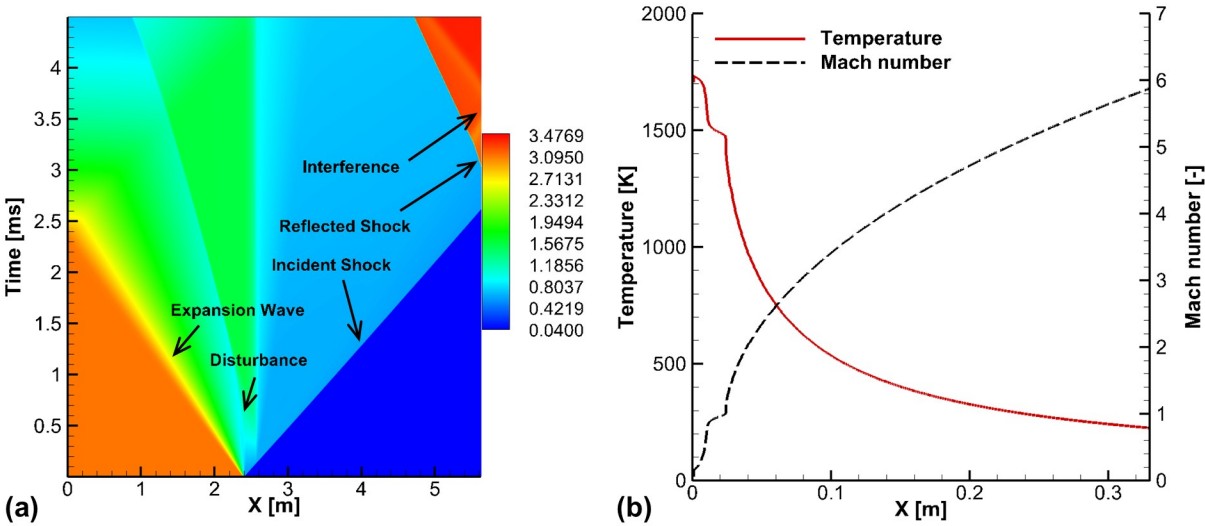

**Fig 5. Calculated results of the shock-tunnel flows.** (a) x-t diagram of pressure in the shock-tube part, (b) Mach number and temperature along the Mach 6 nozzle part.

Fig 5 presents the calculated $x$–$t$ diagram of the pressure in the shock-tube part, and the Mach number and trans-rotational temperature $T_{tr}$ in the nozzle section. In Fig 5(a), it can be observed that the incident shock wave arrives at the end-wall at 2.8 $ms$, and the propagation of the expansion wave started at the diaphragm position. Near 2.50 $m$, the flow is disturbed by the transition section. When the reflected shock wave is produced at the end-wall, the high-pressure state is maintained until it interferes with the following contact surface. The steady-state time of the post-shock condition before the interruption was calculated to be about 1.0 $ms$. The reservoir conditions behind the reflected shock wave were employed as the inlet conditions of the nozzle: $\rho$ = 6.539 $kg/m^3$, $p$ = 3.176 $MPa$, and $T_{tr}$ = 1, 687 $K$. Fig 5(b) presents the Mach number $M$ and trans-rotational temperature $T_{tr}$ in the nozzle expanding section. Fig 5(b) shows that the heated gas expands through the nozzle, while the temperature drops drastically to 260 $K$, and the flow velocity reaches $M$ = 6 at the nozzle exit.

## Thermochemical nonequilibrium analysis

A thermochemical nonequilibrium analysis of the K1 shock-tunnel was conducted using the Q1D flow calculations for the various flow enthalpy conditions. Table 4 summarizes the initial conditions of Cases A to D, used to simulate the low enthalpy flows in the shock-tube and shock-tunnel modes. In the present work, the proposed 3-T model was employed in the Q1D flow calculations to analyze the thermochemical nonequilibrium phenomena and estimate the reservoir and nozzle exit conditions. In addition to the 3-T model, previous 1-T and 2-T models were adopted in the Q1D flow calculations to compare the results, and to investigate the thermochemical nonequilibrium flows. For Cases A, B, and C, the calculated results for the shock-tube and shock-tunnel operation modes were compared with existing experimental data [18, 20, 21]. The shock-tunnel flow calculation was additionally performed for Case D, to numerically investigate the thermochemical nonequilibrium flows in the maximum available dry air gas operating condition.

Fig 6 presents the calculated results for the flow conditions in Case A in the shock-tube operating mode. The shock-tube flow calculation was performed to obtain the thermodynamic properties and oxygen mole-fraction at the end-wall in the shock-tube mode. In the shock-

**Table 4. Initial conditions of low enthalpy shock-tunnel flows for thermochemical nonequilibrium analysis.**

|  | Case A [20] | Case B [21] | Case C [18] | Case D |
|---|---|---|---|---|
| Mode | Shock-tube | Shock-tube | Shock-tunnel | Shock-tunnel |
| Driver gas | He | He | He | He |
| Driven gas | 21% $O_2$-79% Ar | Dry air | Dry air | Dry air |
| $p_4(MPa)$ | 1.65 | 1.60 | 3.1 | 3.1 |
| $p_1(kPa)$ | 1 | 0.25 | 40 | 0.5 |
| $T_4(K)$ | 288 | 288 | 288 | 300 |
| $T_1(K)$ | 288 | 288 | 288 | 300 |
| Channel 1 ($m$) | 3.615 | 3.615 | 5.22 | - |
| Channel 2 ($m$) | 4.115 | 4.115 | 5.72 | - |
| Channel 3 ($m$) | - | - | 5.8 | - |
| End-wall ($m$) | 4.215 | 4.215 | - | - |

tube flow calculation, a test gas of 21% $O_2$ and 79% Ar was adopted, to match the shock-tube experiments performed by Kim et al [20]. Since $O_2$ is a single diatomic species, the calculated results using the 3-T model yielded the same results as the 2-T model. Fig 6(a) presents a comparison of the calculated results and experimental data [20] of the pressure at the end-wall. In the experiments [20], the pressure sensor was mounted at the end-wall, and the reflected shock condition was observed. Before the contact surface arrives, the steady-state conditions remain at about 200 $\mu$s at the end-wall. The pressure calculated using the 3-T model matches the measured pressure well. There was no significant difference in the pressure calculated by the 3-T and 1-T models. After the pressure jump, immediately behind the shock wave, the pressure rapidly falls due to the dissociation reaction of $O_2$. Fig 6(b) presents the calculated results of the temperatures at the end-wall. Due to the endothermic reaction of $O_2$ dissociation, the temperature peak is observed instantaneously after the shock wave arrives. In the species vibrational temperature $T_v$ in the 3-T model, the nonequilibrium phenomena of vibrational relaxation of $O_2$ is rapidly removed after the shock wave, and the 3-T model reaches steady state faster than the 1-T model. In Fig 6(c), the species density at the end-wall is presented. It shows that behind the shock wave the $O_2$ density increases, because the flow is rapidly compressed by the shock wave. Though the trans-rotational temperature $T_{tr}$ in the 3-T model is almost the same as the equilibrium temperature $T$ in the 1-T model, the relaxation of the species vibrational temperature $T_v$ in the 3-T model leads to less pressure $p$ and density $\rho$, and the chemical reactions in the 3-T model appear to be faster than those of the 1-T model, because of the lower $O_2$ density.

Fig 7 presents the calculated results of the shock-tube flows for the flow conditions in Case B. In the present work, the calculated results were compared with the experimental data [21] to validate the 3-T model. In the experiment [21], the pressure sensors were installed in Channel 1, Channel 2, and the end-wall position. The pressures measured in Channels 1 and 2 were used to determine the shock speed, and the pressure at the end-wall was utilized to determine the reservoir conditions of the shock-tube part. The ultraviolet emission spectra of OH radicals in the A-X band range of 280 $nm$ to 330 $nm$ were also measured, at a distance 3 $mm$ from the end-wall. The rotational temperature was determined using the P-Q branch ratio of the measured spectrum. Then, the trans-rotational temperature $T_{tr}$ can be determined by using the rotational temperature.

In Fig 7(a), the calculated pressure at the end-wall is compared with the measured data. Before the contact surface arrives, a steady flow is maintained for 100 $\mu$s. At the initial pressure

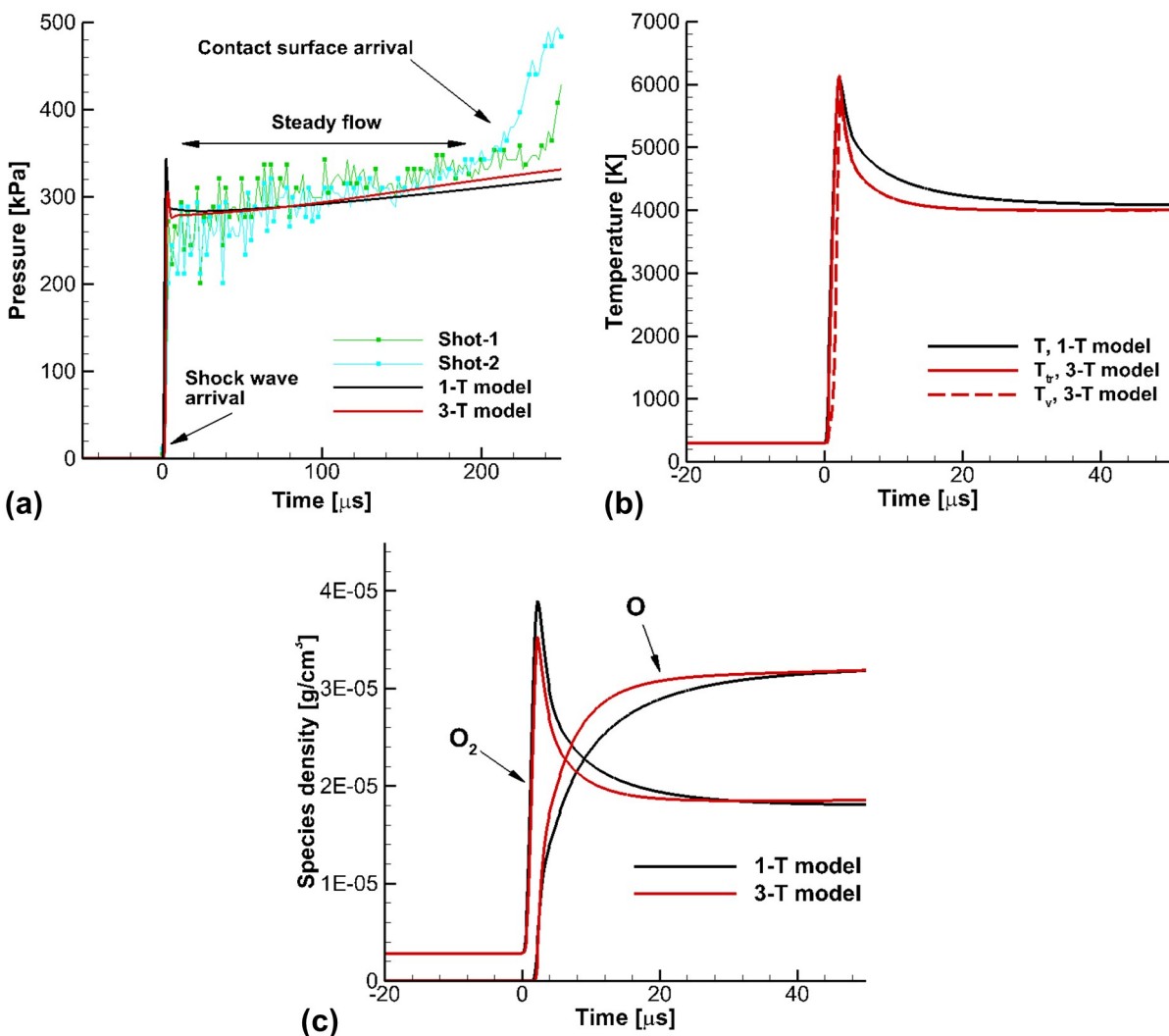

**Fig 6. Results of the shock-tube flows for the flow conditions in Case A.** (a) Pressure at the end wall, (b) Temperatures at the end wall, (c) Species densities at the end wall.

peak, the 3-T model results are closer to the experiment than the results calculated using the 1-T and 2-T models. In Fig 7(b), a comparison of the calculated and experimental temperatures data is presented at the optical sensing point 3 *mm* away from the end-wall. The transrotational temperature $T_{tr}$ calculated by the 3-T model matches the measured temperature well. Compared with the 1-T model, the trans-rotational temperature $T_{tr}$ of the 3-T model is higher than the equilibrium temperature $T$ of the 1-T model after the incident shock arrives. This is because, in the 1-T model, the relaxation process of the nonequilibrium temperatures are neglected. Behind the reflected shock wave, the trans-rotational temperature $T_{tr}$ peak of the 3-T model is much higher than the equilibrium temperature $T$ of the 1-T model, for the same reason. It is observed that there were obvious differences between the nonequilibrium temperatures from the 3-T and 2-T models. The electron-electronic-vibrational temperature of the 3-T model was much lower than that of the 2-T model behind the incident and reflected shock waves. In the 2-T model, the electron-electronic-vibrational relaxations of $N_2$, $O_2$, and NO are treated as a single nonequilibrium relaxation mode. However, in the 3-T model, the

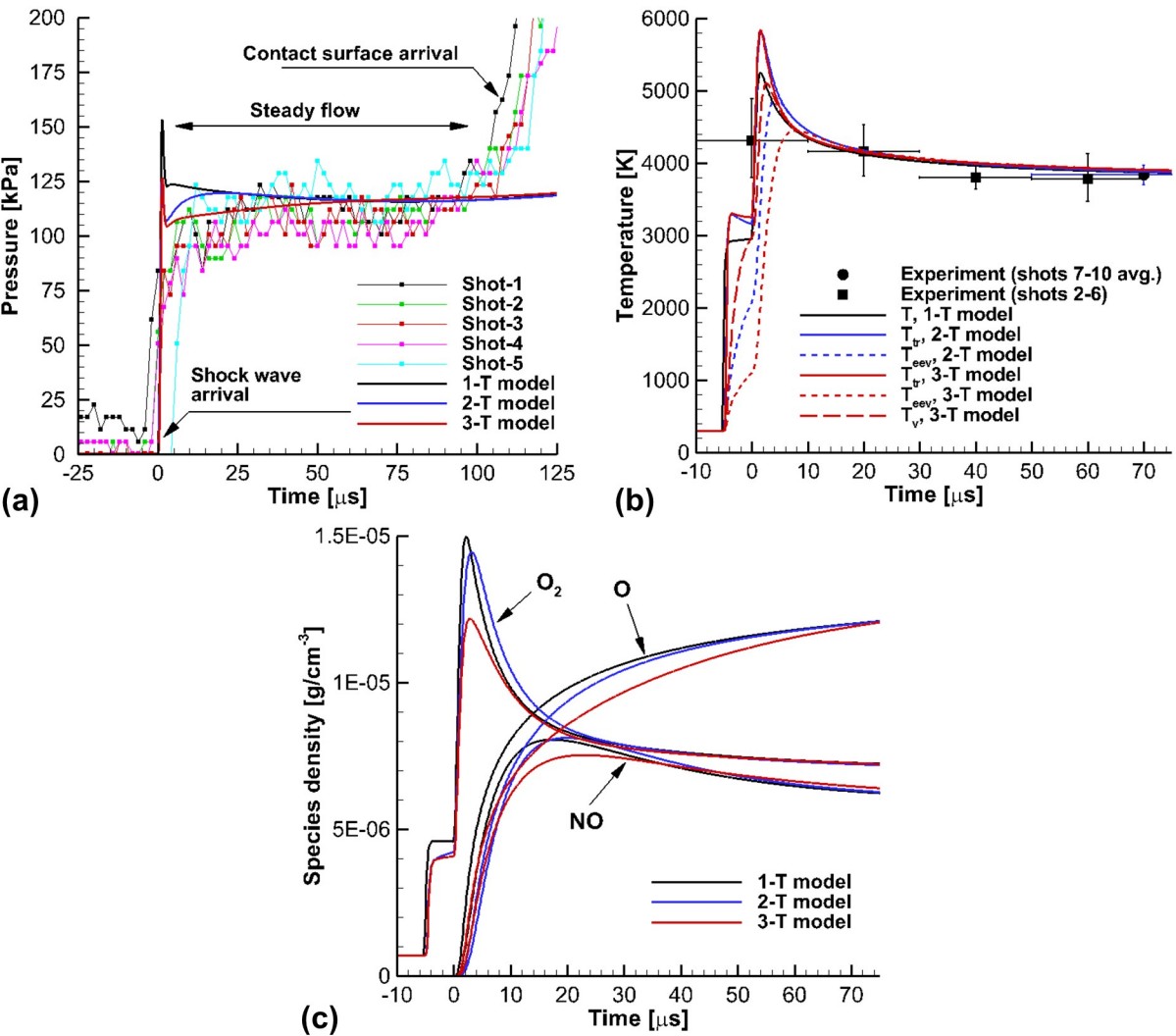

**Fig 7. Results of the shock-tube flows for the flow conditions in Case B.** (a) Pressure at the end wall, (b) Temperatures at the optical sensing point, (c) Species densities at the optical sensing point.

vibrational relaxations of $O_2$ and NO are separated from the single nonequilibrium mode of the 2-T model, and the electron-electronic-vibrational relaxation in the 3-T model is dominantly influenced by the vibrational relaxation of $N_2$. Thus, in the 3-T model, the electron-electronic-vibrational relaxation occurred much more slowly than in the 2-T model. Behind the reflected shock wave, the relaxation time of the electron-electronic-vibrational temperature $T_{eev}$ and the species vibrational temperature $T_v$ in the 3-T model were obviously different. This is because, in the 3-T model, the electron-electronic-vibrational temperature $T_{eev}$ is mainly influenced by the vibrational relaxation of $N_2$, and the vibrational relaxation time of $N_2$ is much longer than that of $O_2$ and NO in low enthalpy flows [17, 28, 29]. This shows that treating the species vibrational temperature $T_v$ of $O_2$ and NO separately from the electron-electronic-vibrational temperature $T_{eev}$ is adequate for describing the thermochemical nonequilibrium of the low-enthalpy flows in the shock-tunnel facility. Fig 7(c) compares the species density at the optical sensing point. In the 3-T model, the dissociation reaction of $O_2$ is controlled by the species vibrational temperature $T_v$, not by the electron-electronic-vibrational

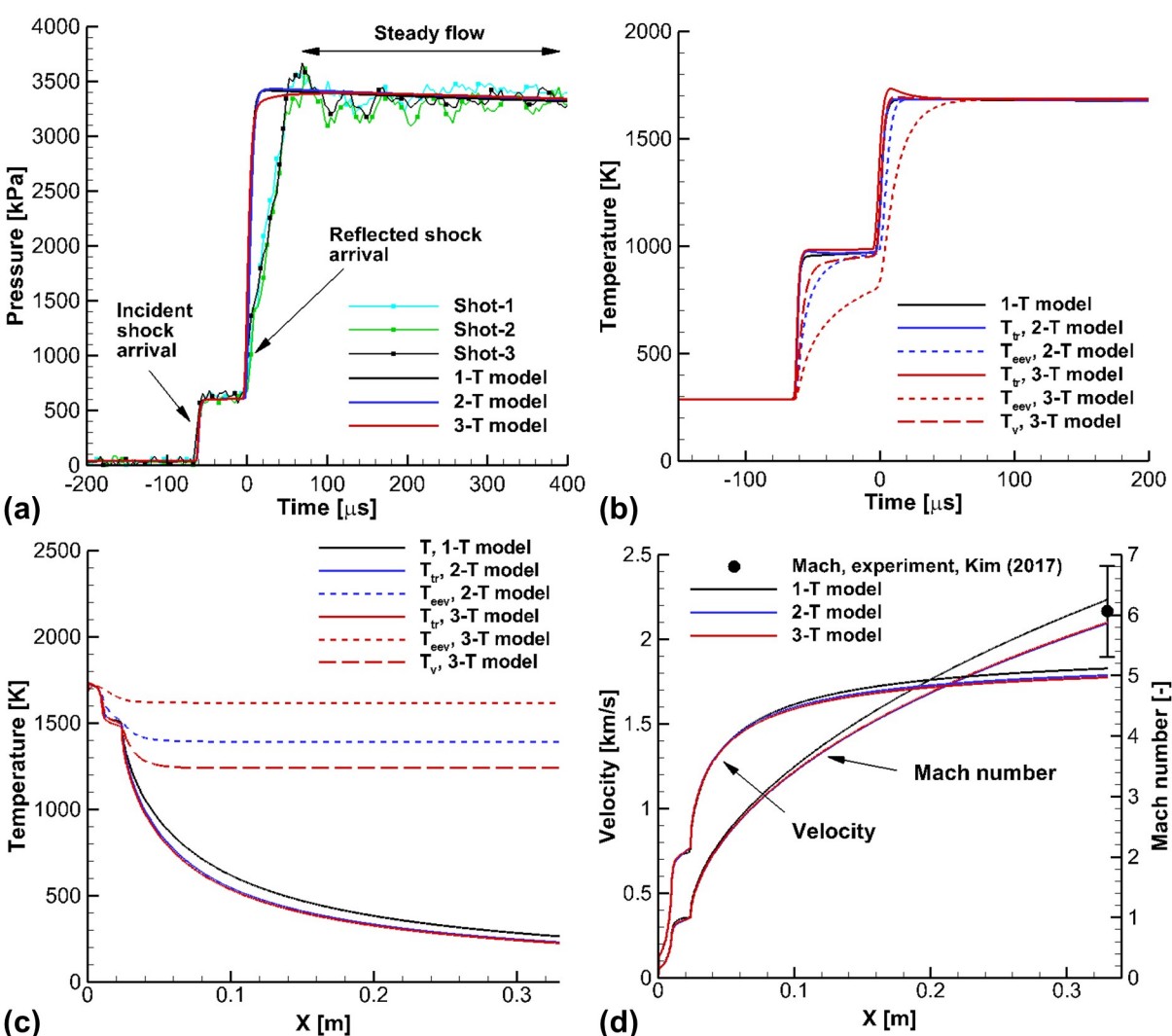

**Fig 8. Results of the shock-tunnel flows for the flow conditions in Case C.** (a) Pressure at the Channel 3, (b) Temperatures at the Channel 3, (c) Temperatures in the nozzle section, (d) Velocity and Mach number in nozzle section.

temperature $T_{eev}$. Thus, the dissociation of $O_2$ is different between the temperature models. As the reaction speed of $O_2$ dissociation determines the amount of O and $O_2$ in the test section within the test time, it could affect the shock-tube test results, e.g. wall ablation and surface catalycity measurement. When comparing the NO species which is mainly produced by $N_2$ and O collisions, the amount of NO in the 3-T model was lower than in the 1-T and 2-T models, because the amount of dissociated O was relatively small in the 3-T model.

Fig 8 shows the calculated results of the shock-tunnel flows for the flow conditions in Case C. The experiments in the shock-tunnel mode were conducted by Kim et al [18]. In Fig 8(a), the calculated pressure is compared with the experimental data which was measured at Channel 3. The pressure jumped two times behind the incident and reflected shock waves, and the 3-T model was able to accurately capture the pressure jump. Differences between the calculated pressure from the 3-T model and the 1-T and 2-T models were negligible. However, there were obvious differences in temperatures. In Fig 8(b), the temperatures calculated by the 3-T model are compared with the temperatures in the 1-T and 2-T models. Behind the

incident shock wave, the electron-electronic-vibrational temperature $T_{eev}$ in the 3-T model exists in a nonequilibrium state when the reflected shock wave arrives. However, the electron-electronic-vibrational temperature $T_{eev}$ in the 2-T model almost converges to the equilibrium values. Behind the reflected shock wave, it was observed that the nonequilibrium state of the 3-T model lasts longer than in the 2-T model. In the flow conditions of Case C, the chemical reactions behind the reflected shock wave are ignorable, because the reservoir temperature and enthalpy are low, at 1,686 $K$ and 1.87 $MJ/kg$, respectively, and this is not enough to produce the chemical reaction. Fig 8(c) shows the calculated temperatures in the Mach 6 nozzle. In the nozzle calculations, the reservoir conditions behind the reflected shock wave were employed as the inlet conditions of the nozzle. In the nozzle, the heated gas is accelerated in the expanding section, while the nonequilibrium temperatures and chemical reactions are rapidly frozen. Comparing the nonequilibrium temperatures in the 3-T and 2-T models, it was observed that the frozen electron-electronic-vibrational temperature $T_{eev}$ in the 3-T model was more highly distributed than in the 2-T model. In addition, the species vibrational temperature $T_v$ in the 3-T model was obviously different from the electron-electronic-vibrational temperature $T_{eev}$ in the 2-T model, due to the different vibrational relaxation times of $N_2$ and $O_2$ in low temperatures below 2,000 $K$ [17, 28, 29]. Due to the freezing of nonequilibrium temperatures in 2-T and 3-T models, it is observed that the trans-rotational temperatures $T_{tr}$ in the 2-T and 3-T models are higher that that that in the 1-T model.

In Fig 8(d), the calculated velocity and Mach number are presented. In the experiments by Kim et al. [18], the shock stand-off distance of a spherical model was measured using the Z-type shadowgraph method. From the ratio of the shock stand-off distance and the radius of the sphere model, the Mach number can be the obtained using the theoretical model. It is observed that the Mach number calculated with the 3-T model reached 6 at the nozzle exit, and this agrees well with the experimental value. The measured Mach number was also reproduced well in the 1-T and 2-T models. From the calculated results for the velocity, it is observed that the velocity in the 3-T model is lower than those of 1-T and 2-T models. In the 3-T model, the electron-electronic-vibrational temperature $T_{eev}$, which includes the vibrational temperature of $N_2$, rapidly freezes. As the $N_2$ is the major species of dry air, the energy locked in nonequilibrium state of electron-electronic-vibrational temperature $T_{eev}$ is not negligible, and this energy is unavailable as kinetic energy [54]. Consequently, the velocity in the 3-T model is the lowest.

Fig 9 shows the calculated results for the flow conditions of Case D. Case D has the maximum operating conditions in the K1 shock-tunnel facility with dry air. In Fig 9(a), the temperatures calculated using the 3-T model are compared with the temperatures obtained with the 1-T and 2-T models at the location of Channel 3. Discernable differences are observed in the nonequilibrium temperatures of the 3-T and 2-T models. The electron-electronic-vibrational temperature $T_{eev}$ in the 3-T model relaxes slower than in the 2-T model behind the incident and reflected shock waves. Before the contact surface arrival, the flows maintain a steady state for 200 $\mu s$. In Fig 9(b), the calculated species densities are represented at the location of Channel 3. After the incident and reflected shock wave arrive, the species densities jump drastically, due to shock wave compression. $O_2$ is dissociated, and the exchange reaction occurs due to collisions between $N_2$ and dissociated O behind the reflected shock wave. The differences in the species densities in the 3-T model and 1-T and 2-T models were negligible. Fig 9(c) shows the calculated temperatures in the Mach 6 nozzle. The frozen electron-electronic-vibrational temperature $T_{eev}$ in the 3-T model is drastically higher than the 2-T model. The difference in the species vibration temperature $T_v$ and the electron-electronic-vibrational temperature $T_{eev}$ in the 3-T model is obvious. These results show that, in order to describe the nonequilibrium free stream conditions at the nozzle exit, the vibrational temperatures of $O_2$ and NO need to be separated from the electron-electronic-vibrational temperature $T_{eev}$. The results also

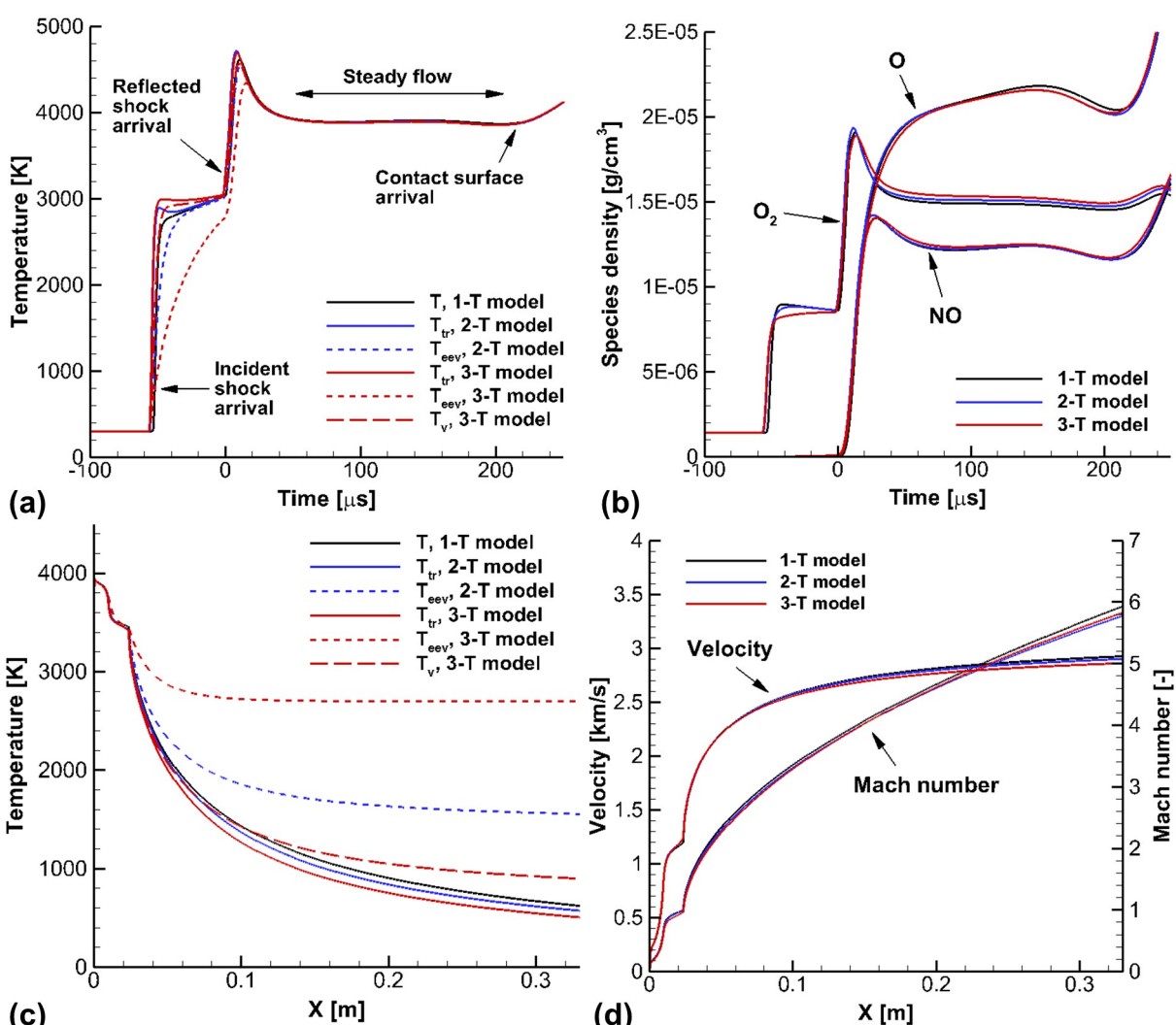

**Fig 9. Results of the shock-tunnel flows for the flow conditions in Case D.** (a) Pressure at the Channel 3, (b) Temperatures at the Channel 3, (c) Temperatures in the nozzle section, (d) Velocity and Mach number in the nozzle section.

indicate the 3-T model can more accurately capture the nonequilibrium state than the 1-T and 2-T models. Fig 9(d) shows the calculated velocity and Mach number in the nozzle. At the nozzle exit, the velocity calculated using the 3-T model is slightly lower than the values calculated by the 1-T and 2-T models. The Mach numbers in 3-T and 2-T models are slightly lower than the designed Mach number.

In Table 5, the results of the low enthalpy shock-tube and shock-tunnel flows are summarized, showing the calculated results from the 3-T model, and the experimental data [18, 20, 21] for the flow conditions for Cases A to D. In the shock-tube flows for Case A and Case B, the calculated reservoir pressure $p_5$ agrees with the measured values, and the shock speed is also accurately reproduced by the 3-T model. In the shock-tunnel flows for Case C and Case D, the measured values of the flow enthalpy and Mach number are accurately reproduced by the 3-T model. In the present work, the modified Rayleigh pitot formula [55] was adopted to convert the flow properties to the total pressure $p_0$. The modified Rayleigh pitot formula is

**Table 5. Reservoir and nozzle exit conditions of K1 shock-tunnel flows.**

| | Case A | | Case B | | Case C | | Case D |
|---|---|---|---|---|---|---|---|
| | Exp. [20] | Calc. | Exp. [21] | Calc. | Exp. [18] | Calc. | Calc. |
| *Shock-tube part* | | | | | | | |
| $p_2$ (kPa) | 52.7±1.9 | 51.3 | 14.2±0.8 | 16.7 | 616.7±18.3 | 601.5 | 29.8 |
| $p_5$ (kPa) | 276±3.0 | 279 | 110±6.9 | 114 | 3344±89.8 | 3391 | 221 |
| $T_5$ (K) | - | 4011 | 3750±215 | 3938 | - | 1686 | 3893 |
| $h_5$ (MJ/kg) | - | 4.23 | - | 6.13 | - | 1.87 | 7.32 |
| $u_{shock}$ (km/s) | 1.96±0.5 | 2.01 | 2.50±0.3 | 2.52 | 1.15±0.02 | 1.19 | 2.44 |
| *Nozzle part* | | | | | | | |
| $h_e$ (MJ/kg) | - | 4.61 | - | 7.56 | 1.68±0.04 | 1.64 | 7.21 |
| $M_\infty$ (−) | - | 7.90 | - | 6.06 | 6.06±0.75 | 5.87 | 5.822 |
| $p_{pitot}$ (kPa) | - | 5.62 | - | 2.50 | 85.0±1.69 | 80.7 | 2.648 |
| $T_{tr}$ (K) | - | 209 | - | 462 | - | 225 | 507 |
| $T_{eev}$ (K) | - | 4006 | - | 2940 | - | 1610 | 2949 |
| $T_v$ (K) | - | 619 | - | 966 | - | 1238 | 1033 |

given as

$$p_0 = p_\infty + \frac{1}{2}\rho_\infty u_\infty^2 \left( \frac{\gamma+1}{\gamma} \left( \frac{(\gamma+1)^2 M_\infty^2}{4\gamma M_\infty^2 - 2\gamma + 2} \right)^{\frac{1}{\gamma-1}} - \frac{2}{\gamma M_\infty^2} \right), \tag{37}$$

where $\gamma$ is specific heat ratio. Then, the calculated total pressure from the 3-T model is found to be in good agreement with the measured pitot pressure at the nozzle exit, for the flow conditions in Case C. In the flow properties at the nozzle exit for Case C and Case D, a strong nonequilibrium state exists, and a significant difference between the electron-electronic-vibrational temperature $T_{eev}$ and species vibrational temperature $T_v$ is observed. Also, the difference between the nonequilibrium temperatures increased proportionally to the flow enthalpy of the nozzle exit. In Case A, the Mach number at the nozzle exit is higher than the designed Mach number 6, due to the different gas composition other than dry air.

## Discussion and conclusions

In the present work, a thermochemical nonequilibrium analysis of the K1 shock-tunnel facility was performed, for low enthalpy operating conditions below 8 *MJ/kg*. To accurately describe the thermochemical nonequilibrium state in low enthalpy flows, a three-temperature (3-T) model is proposed. In the 3-T model, the electron translational energy, the electronic energies of atoms and molecules, and the vibrational energy of $N_2$ are grouped as one nonequilibrium energy mode of the electron-electronic-vibrational energy, and the vibrational energies of $O_2$, NO, and $O_2^+$ are treated as another nonequilibrium energy mode of the species vibrational energy. In the shock-tunnel flow calculation, a quasi-one-dimensional method was employed by dividing the flow calculations into two-parts, consisting of the shock-tube and the Mach 6 nozzle. The calculated results of the shock-tunnel flows obtained from the 3-T model were then compared with existing experimental data, and the calculated results obtained from previous one-temperature (1-T) and two-temperature (2-T) models, for various operating conditions in the K1 shock-tunnel facility.

In the thermochemical nonequilibrium analysis in low enthalpy shock-tunnel flows, discernible differences were observed in the nonequilibrium temperatures behind the incident

and reflected shock waves. The electron-electronic-vibrational temperature of the 3-T model relaxes more slowly to the equilibrium temperature than the 2-T model behind the incident and reflected shock waves. Depending on the flow conditions, the relaxation time for electron-electronic-vibrational temperature of the 3-T model is 1.25 to 3 times slower than that of the 2-T model. This is because, in the 3-T model, the electron-electronic-vibrational temperature is mainly influenced by the vibrational relaxation of $N_2$, and the vibrational relaxation time of $N_2$ is much slower than that of $O_2$ and NO in low enthalpy flows. In the chemical reactions behind the incident and reflected shock waves, $O_2$ dissociation is dominant, and the reaction speed predicted by the 3-T and 2-T models is different. The differences of nonequilibrium characteristics of $N_2$ and $O_2$ are also obviously apparent in the nozzle expanding flows where the vibrational temperature freezes. In 3-T model, depending on the flow condition, the frozen electron-electronic-vibrational temperature at the nozzle exit is 15–75% higher than that in the 2-T model. On the other hand, the species vibrational temperature of the 3-T model is around 10–40% lower than the electron-electronic-vibrational temperature in the 2-T model. Due to the rapid freezing of electron-electronic-vibrational temperature in the 3-T model, some part of energy is locked as nonequilibrium energies so that the trans-rotational temperature and velocity in the 3-T model are lower than those of 1-T and 2-T models.

From the thermochemical nonequilibrium analysis of low enthalpy shock-tunnel flows, it is suggested that the nonequilibrium characteristics of $N_2$ and $O_2$ need to be treated separately. The present 3-T model was demonstrated capable of describing the nonequilibrium characteristics behind the incident and reflected shock waves and at the nozzle exit.

## Author Contributions

**Conceptualization:** Sanghoon Lee.

**Data curation:** Jae Gang Kim.

**Funding acquisition:** Jong Kook Lee.

**Investigation:** Ikhyun Kim.

**Methodology:** Sanghoon Lee.

**Project administration:** Gisu Park.

**Software:** Sanghoon Lee, Jae Gang Kim.

**Supervision:** Jae Gang Kim.

**Writing – original draft:** Sanghoon Lee.

**Writing – review & editing:** Jae Gang Kim.

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
