## [Decision Letter · Decision Letter 0]

10 Aug 2020

PONE-D-20-06935

Thermochemical nonequilibrium flow analysis in low enthalpy shock-tunnel facility

PLOS ONE

Dear Dr. Kim,

Thank you for submitting your manuscript to PLOS ONE. After careful consideration, we feel that it has merit but does not fully meet PLOS ONE’s publication criteria as it currently stands. Therefore, we invite you to submit a revised version of the manuscript that addresses the points raised during the review process.

We look forward to receiving your revised manuscript.

Kind regards,

Hongbing Ding, Ph.D.

Academic Editor

PLOS ONE

Additional Editor Comments:

Thank you for submitting your manuscript to PLOS ONE. The reviewers recommend reconsideration of your paper following major revision. I invite you to resubmit your manuscript after addressing all reviewer comments.

Journal Requirements:

Reviewers' comments:

Reviewer's Responses to Questions

**Comments to the Author**

1. Is the manuscript technically sound, and do the data support the conclusions?

Reviewer #1: Partly

Reviewer #2: Yes

2. Has the statistical analysis been performed appropriately and rigorously? 

Reviewer #1: I Don't Know

Reviewer #2: N/A

3. Have the authors made all data underlying the findings in their manuscript fully available?

Reviewer #1: Yes

Reviewer #2: Yes

4. Is the manuscript presented in an intelligible fashion and written in standard English?

Reviewer #1: Yes

Reviewer #2: Yes

5. Review Comments to the Author

Reviewer #1: 1. the flow which is treated as quasi-one-dimensional flow is right or not?

2. the three-temperature model is proposed by author or referenced other authors?

3. treating the vibrational nonequilibrium of O2 and NO separately is not suitable.

4. the conclusions should include more information, data is better.

Reviewer #2: Overall Comments

-This is a good paper for this journal and should be accepted with some revisions.

-A very detailed analysis, but in real shock tube flows, the three-dimensional effects are going to swamp the thermochemical issues associated with the two-temperature model.

Abstract

The proposed three-temperature model was able to describe the different thermochemical nonequilibrium characteristics of N2 and O2 behind the incident and reflected shock waves and at the nozzle exit.

-I’m not sure what is meant by different. Please clarify specific fluid mechanic of thermochemical phenomena

Introduction

Unfortunately, it is difficult to measure all flow properties, because the flow duration is typically only a few hundreds of milliseconds.

-You mean a few milliseconds. A few hundred would be terrific.

Darcy-Weisbach friction factor f

-This assumes laminar flow, correct? This may not be the case.

Na was employed to measure the electron-electronic-vibrational temperature Teev. The electronic temperature was modeled on the emissions spectra of Na, and the concentration of Na was varied between 10 and 100 ppm by volume.

-Seeding sodium in the shock tunnel is going to change the reaction rates. Why is this a valid comparison?

Reservoir and nozzle exit conditions of K1 shock-tunnel flows.

-Why are only some conditions completely listed. Please list all conditions.

6. PLOS authors have the option to publish the peer review history of their article (what does this mean?). If published, this will include your full peer review and any attached files.

Reviewer #1: No

Reviewer #2: No

---

## [Author Response · Author response to Decision Letter 0]

15 Sep 2020

Reply to the Reviewer’s Comments

Manuscript ID: PONE-D-20-06935

Type: Research Article

Title: Thermochemical nonequilibrium flow analysis in low enthalpy shock-tunnel facility 

Authors: Sanghoon Lee, Ikhyun Kim, Gisu Park (Korea Advanced Institute of Science and Technology), Jong Kook Lee (Agency for Defense Development), and Jae Gang Kim (Sejong University; Corresponding author)

We would like to express our sincere appreciation for the reviewers’ valuable comments and suggestions on the manuscript. We have addressed all the comments and suggestions raised. We hope that the revised manuscript is now acceptable by the reviewers.

 

Reviewer #1

Comment 1:

The flow which is treated as quasi-one-dimensional flow is right or not?

Response:

The three-dimensional flow calculations have an advantage of the performance analysis of the shock-tunnel facility1. However, the computational costs of the three-dimensional calculations are enormous to analyze the thermochemical nonequilibrium phenomena of the shock-tunnel flows. In the work by Nomelis et al.2, the two-dimensional axi-symmetric calculations of the shock-tunnel flow were preformed, and it was compared with the results by the quasi-one-dimensional calculations and the measured data of the shock-tunnel facility. The comparisons confirmed that the quasi-one-dimensional calculations are acceptable to analyze the thermochemical nonequilibrium of the shock-tunnel flows. Hence, the thermochemical nonequilibrium analysis of the shock-tunnel flows has been widely performed by using the quasi-one-dimensional calculations3. 

In the revised manuscript, the explanation of the quasi-one-dimensional calculations are added in ‘Introduction’ on the lines 28-41 in page 3. 

[1] Luo K, Wang Q, Li J, Li J, Zhao W. Numerical nodeling of a high-enthalpy shock tunnel driven by gaseous detonation. Aerosp Sci Technol. 2020;104. doi:10.1016/j.ast.2020.105958.

[2] Nompelis I. Computational study of hypersonic double-cone experiments for code validation. Department of Aerospace Engineering and Mechanics, University of Minnesota, Minnesota, USA: Ph.D. Dissertation; 2004. 

[3] Jacobs PA. Quasi-one-dimensional modeling of a free-piston shock tunnel. AIAA J. 1994;32(1):137{145. doi:10.2514/3.11961.

Comment 2:

The three-temperature model is proposed by author or referenced other authors?

Response:

The three-temperature model is proposed by authors. There have been several attempts to model the thermochemical nonequilibrium phenomena using a multi-temperature model. In the present three-temperature model, unlike the previous thermochemical nonequilibrium models, the electron-electronic-N2 vibrational temperature and species vibrational temperature of O2 and NO are treated as the separated nonequilibrium modes. 

Comment 3:

Treating the vibrational nonequilibrium of O2 and NO separately is not suitable

Response:

As a reviewer commented, in the previous two-temperature model, the vibrational nonequilibrium modes of O2 and NO do not be treated as separately. The objective of the previous two-temperature model is to describe the nonequilibrium phenomena at the high-enthalpy flows above 8 MJ/kg. In the high enthalpy environments, the dissociation of O2 occurs rapidly, and the influence of the species vibrational nonequilibrium of O2 and NO is not significant. However, in low-enthalpy shock tunnel flows below 8 MJ/kg, where the stagnation temperature is less than 6,000 K, the dissociation rate of O2 is mild, and it couples with the species vibrational nonequilibrium of O2. Therefore, a more sophisticated consideration of species vibrational nonequilibrium is required. 

In the low-enthalpy shock tunnel flows, the species vibrational relaxation time of O2 differs from that of N2, and the species vibrational nonequilibrium of NO is not significant because the amount of NO produced from the exchange reaction is relatively small. Hence, in the present work, the species vibrational nonequilibrium of O2 and NO are treated as one species vibrational nonequilibrium mode, and treated separately from the electron-electronic-N2 vibrational nonequilibrium mode. 

Comment 4:

The conclusions should include more information, data is better.

Response:

As the reviewer suggested. We add more information which can be inferred from the present work in ‘Discussion and conclusions’ on the lines 565-581 in pages 25-26. 

Reviewer #2

Overall comments:

This is a good paper for this journal and should be accepted with some revisions. A very detailed analysis, but in real shock tube flows, the three-dimensional effects are going to swamp the thermochemical issues associated with the two-temperature model.

Response:

As a reviewer commented, there are three-dimensional effects in real shock-tube which significantly affect the flow, especially in terms of performance and measurement aspects. In the present work, we limited our scope to quasi-one-dimensional flow for the detailed analysis of shock-tunnel flows in thermochemical nonequilibrium aspects. 

In the revised manuscript, the more explanation of the quasi-one-dimensional calculations are added in ‘Introduction’ on the lines 28-41 in page 3. 

Comment 1 (Abstract):

‘The proposed three-temperature model was able to describe the different thermochemical nonequilibrium characteristics of N2 and O2 behind the incident and reflected shock waves and at the nozzle exit’. I’m not sure what is meant by different. Please clarify specific fluid mechanic of thermochemical phenomena.

Response:

As the reviewer suggested, we added the physical explanation of the present results in ‘Abstract’.

Comment 2 (Introduction):

‘Unfortunately, it is difficult to measure all flow properties, because the flow duration is typically only a few hundreds of milliseconds. You mean s few milliseconds. A few hundred would be terrific.

Response:

It is our mistake to express as “a few hundreds of milliseconds”. As the reviewer suggested, we corrected the sentence on the lines 24-25 in page 3. 

Comment 3 (Thermochemical nonequilibrium model):

Darcy-Weisbach friction factor f. This assumes laminar flow, correct? This may not be the case.

Response:

We believe that Darcy-Weisbach friction factor f can be used in the shock-tube flow where the Reynolds number is up to 〖10〗^5. Though the turbulent flow cannot be directly considered in quasi-one-dimensional calculation, the viscous effects can be considered. This can be achieved by calculating the wall shear stress in the viscous source term using Darcy-Weisbach friction factor f, which can be explicitly determined from Reynolds number. The correlation of Reynolds number and Darcy-Weisbach friction factor f has been extensively investigated in the Reynolds number range of 3000 to 〖10〗^8. This method has been generally adopted for quasi-one-dimensional calculation of shock-tunnel flows, for example, L1D code in the university of Queensland. We reflected this comment in the revised manuscript including the equation on the lines 258-263 in pages 13-14. 

Comment 4 (Thermochemical nonequilibrium analysis in the K1 shock-tunnel facility):

‘Na was employed to measure the electron-electronic-vibrational temperature T_eev. The electronic temperature was modeled on the emissions spectra of Na, and the concentration of Na was varied between 10 and 100 ppm by volume’. Seeding sodium in the shock tunnel is going to change the reaction rates. Why is this a valid comparison?

Response:

We agree with the reviewer’s comment. In line-reversal methods, the metallic seeding species are used to approximately measure the vibrational temperature of diatomic molecules. However, it is found that the concentration of seeding species could enhance the vibrational relaxation process. In the Park’s work, which this study benchmarked, the vibrational relaxation time between Na and N2 was set to physically-fastest time to reflect this matter, and the calculation for the nozzle flow was performed under varying concentration of Na up to 100 ppm. The present study followed this approach, and it can be seen in Fig 3 (b) that the N2 vibrational temperature changes with the concentration of Na. For conciseness, we have only mentioned about the method used in Park’s work, but have not included the results of the Park’s work. We reflected this matter by modifying the Fig 3 (b) which now includes the calculation results from Park’s work, for the code-to-code validation as well as the comparison with experimental measurements on the lines 302-313 in page 16. 

Comment 5 (Thermochemical nonequilibrium analysis in the K1 shock-tunnel facility):

‘Reservoir and nozzle exit conditions of K1 shock-tunnel flows’. Why are only some conditions completely listed. Please list all conditions. 

Response:

As a reviewer suggested, we corrected the Table 5 in page 25.

---

## [Decision Letter · Decision Letter 1]

18 Sep 2020

PONE-D-20-06935R1

Thermochemical nonequilibrium flow analysis in low enthalpy shock-tunnel facility

PLOS ONE

Dear Dr. Kim,

Thank you for submitting your manuscript to PLOS ONE. After careful consideration, we feel that it has merit but does not fully meet PLOS ONE’s publication criteria as it currently stands. Therefore, we invite you to submit a revised version of the manuscript that addresses the points raised during the review process.

We look forward to receiving your revised manuscript.

Kind regards,

Hongbing Ding, Ph.D.

Academic Editor

PLOS ONE

Additional Editor Comments (if provided):

The reviewers recommend reconsideration of your paper following minor revision. I invite you to resubmit your manuscript after addressing all reviewer comments.

Reviewers' comments:

Reviewer's Responses to Questions

**Comments to the Author**

1. If the authors have adequately addressed your comments raised in a previous round of review and you feel that this manuscript is now acceptable for publication, you may indicate that here to bypass the “Comments to the Author” section, enter your conflict of interest statement in the “Confidential to Editor” section, and submit your "Accept" recommendation.

Reviewer #1: (No Response)

Reviewer #2: All comments have been addressed

2. Is the manuscript technically sound, and do the data support the conclusions?

Reviewer #1: Partly

Reviewer #2: Yes

3. Has the statistical analysis been performed appropriately and rigorously? 

Reviewer #1: Yes

Reviewer #2: Yes

4. Have the authors made all data underlying the findings in their manuscript fully available?

Reviewer #1: Yes

Reviewer #2: Yes

5. Is the manuscript presented in an intelligible fashion and written in standard English?

Reviewer #1: Yes

Reviewer #2: Yes

6. Review Comments to the Author

Reviewer #1: The authors have send the revised version of their article. The quality of the article has improved but some of the remarks haven't been addressed in a satisfactory level.

The introduction is still superficial. The reviewer suggests the author should refer some works about non-equilibrium condensation model, such as: [International Journal of Multiphase Flow,2019(114)180-191.], [Energy Conversion & Management, 2019, 181:159–177.], [DOI:10.1016/j.energy.2019.115982. ], [doi: https://doi.org/10.1016/j.applthermaleng.2019.114388.], and so on.

Besides, the reason of non-equilibrium flow should be explained, the reference maybe could help you [Applied Thermal Engineering. 2020;171:115090.],[https://doi.org/10.1016/j.energy.2020.118690.],[DOI: 10.1016/j.ijmultiphaseflow.2019.103083.].

Reviewer #2: All of the comments that I originally made were addressed. All of the comments that I originally made were addressed.

7. PLOS authors have the option to publish the peer review history of their article (what does this mean?). If published, this will include your full peer review and any attached files.

Reviewer #1: No

Reviewer #2: No

---

## [Author Response · Author response to Decision Letter 1]

21 Sep 2020

Reviewer #1

Comment 1:

The authors have send the revised version of their article. The quality of the article has improved but some of the remarks haven't been addressed in a satisfactory level.

The introduction is still superficial. The reviewer suggests the author should refer some works about non-equilibrium condensation model, such as: 

[International Journal of Multiphase Flow, 2019(114)180-191.], 

[Energy Conversion & Management, 2019, 181:159-177.],

[doi:10.1016/j.applthermaleng.2019.114388.], and so on. 

Besides, the reason of non-equilibrium flow should be explained, the reference maybe could help you

[Applied Thermal Engineering. 2020;171:115090.], 

[doi:10.1016/j.energy.2020.118690.],

[doi:10.1016/j.ijmultiphaseflow.2019.103083.].

Response:

 We appreciate the reviewer’s comment for broadening our knowledge. As the reviewer suggested, we modified on lines in ‘Introduction’ on the lines 23-31 in pages 2-3.

---

## [Editor Report · Decision Letter 2]

24 Sep 2020

Thermochemical nonequilibrium flow analysis in low enthalpy shock-tunnel facility

PONE-D-20-06935R2

Dear Dr. Kim,

We’re pleased to inform you that your manuscript has been judged scientifically suitable for publication and will be formally accepted for publication once it meets all outstanding technical requirements.

Kind regards,

Hongbing Ding, Ph.D.

Academic Editor

PLOS ONE

Additional Editor Comments (optional):

The authors have done a good job in revising the manuscript. Now it can be accepted for publication in PLOS ONE.
---

## [Editor Report · Acceptance letter]

28 Sep 2020

PONE-D-20-06935R2 

Thermochemical nonequilibrium flow analysis in low enthalpy shock-tunnel facility 

Dear Dr. Kim:

I'm pleased to inform you that your manuscript has been deemed suitable for publication in PLOS ONE. Congratulations! Your manuscript is now with our production department. 

Kind regards, 

on behalf of

Professor Hongbing Ding 

Academic Editor

PLOS ONE